# Learning Distributions on Manifolds
# with Free-Form Flows

**Peter Sorrenson\*, Felix Draxler\*, Armand Rousselot\*, Sander Hummerich, Ullrich Köthe**
Computer Vision and Learning Lab, Heidelberg University
\*equal contribution, `firstname.lastname@iwr.uni-heidelberg.de`

## Abstract

We propose Manifold Free-Form Flows (M-FFF), a simple new generative model for data on manifolds. The existing approaches to learning a distribution on arbitrary manifolds are expensive at inference time, since sampling requires solving a differential equation. Our method overcomes this limitation by sampling in a single function evaluation. The key innovation is to optimize a neural network via maximum likelihood on the manifold, possible by adapting the free-form flow framework to Riemannian manifolds. M-FFF is straightforwardly adapted to any manifold with a known projection. It consistently matches or outperforms previous single-step methods specialized to specific manifolds. It is typically two orders of magnitude faster than multi-step methods based on diffusion or flow matching, achieving better likelihoods in several experiments. We provide our code at `https://github.com/vislearn/FFF`.

## 1 Introduction

Generative models have achieved remarkable success in various domains such as image synthesis [Rombach et al., 2022], natural language processing [Brown et al., 2020], scientific applications [Noé et al., 2019] and more. However, the approaches are not directly applicable when dealing with data inherently structured in non-Euclidean spaces, which is common in fields such as the natural sciences, computer vision, and robotics. Examples include earth science data on a sphere, the orientation of real-world objects given as a rotation matrix in $SO(3)$, or data on special geometries modeled by meshes or signed distance functions. Representing such data naively using internal coordinates, such as angles, can lead to topological issues, causing discontinuities or artifacts.

Luckily, many generative models can be adapted to handle data on arbitrary manifolds. However, the predominant methods compatible with arbitrary Riemannian manifolds involve solving differential equations—stochastic (SDEs) or ordinary (ODEs)—for sampling and density estimation [Rozen et al., 2021, Mathieu and Nickel, 2020, Huang et al., 2022, De Bortoli et al., 2022, Chen and Lipman, 2024]. These methods are computationally intensive due to the need for numerous function evaluations during integration, slowing down inference.

To address these challenges, we introduce a novel approach for modeling distributions on arbitrary Riemannian manifolds that circumvents the computational burden of previous methods. This is achieved by using a single feed-forward neural network on an embedding space as a generator, with outputs projected to the manifold (Fig. 1). We learn this network as a normalizing flow, facilitated by generalizing the free-form flow framework [Draxler et al., 2024, Sorrenson et al., 2024] to Riemannian manifolds. The core innovation is estimating the gradient of the negative log-likelihood within the tangent space of the manifold.

In particular, we make the following contributions:

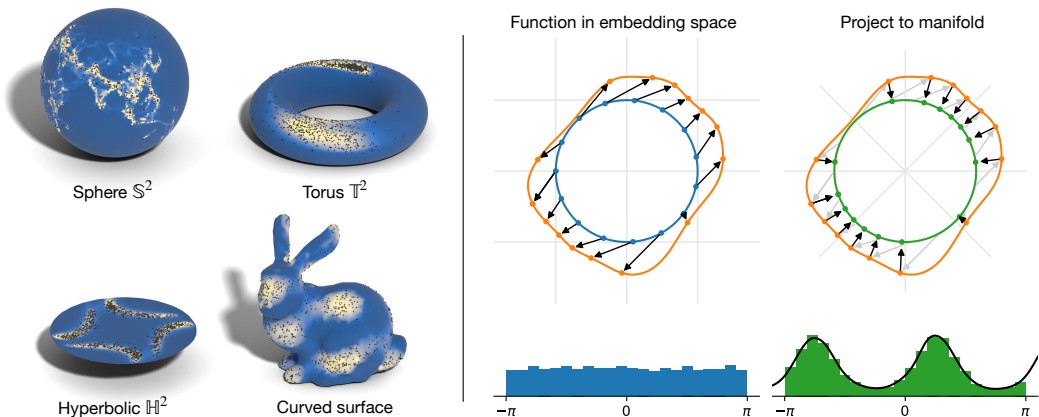

Figure 1: Manifold Free-Form Flows (M-FFF) learn generative models on a variety of manifolds. *(Left)* The learned distributions *(colored surface)* accurately match the test points *(black dots)*. *(Right)* We parameterize M-FFF using a neural network in an embedding space, whose outputs are projected to the manifold. This enables simulation-free training and inference, and naturally respects the corresponding geometry, yielding fast sampling and continuous distributions regardless of the manifold.

- We extend free-form flows to Riemannian manifolds, yielding manifold free-form flows (M-FFF) in Section 4.
- M-FFF can easily be adapted to arbitrary Riemannian manifolds, requiring only a projection function from an embedding space.
- It only relies on a single function evaluation during training and sampling, speeding up inference over multi-step methods typically by two orders of magnitude.
- M-FFF consistently matches or outperforms previous single-step methods on several benchmarks on spheres, tori, rotation matrices, hyperbolic space and curved surfaces (see Fig. 1 and Section 5). In addition, it is consistently faster than multi-step methods by two orders of magnitude, while also outperforming them in terms of likelihood in several cases.

Together, manifold free-form flows offer a novel and efficient approach for learning distributions on manifolds, applicable to any Riemannian manifold with a known embedding and projection.

## 2 Related work

Table 1: Feature comparison of generative models on manifolds. We give a "✓" if any method in a category meets this requirement.

|  | Respects topology | Single step sampling | Arbitrary manifolds |
| --- | --- | --- | --- |
| Euclidean | ✗ | ✓ | ✓ |
| Specialized | ✓ | ✓ | ✗ |
| Continuous time | ✓ | ✗ | ✓ |
| M-FFF (ours) | ✓ | ✓ | ✓ |

Existing work on learning distributions on manifolds can be broadly categorized as follows: (i) leveraging Euclidean generative models; (ii) building specialized architectures that respect one particular kind of geometry; and (iii) learning a continuous time process on the manifold. We compare our method to these approaches in Table 1 and give additional detail below.

**Euclidean generative models.** One approach maps the $n$-dimensional manifold to $\mathbb{R}^n$ and learns the resulting distribution [Gemici et al., 2016]. Another approach generalizes the reparameterization trick to Lie groups by sampling on the Lie algebra which can be parameterized in Euclidean space [Falorsi et al., 2019]. These approaches come with the downside that a Euclidean representation

may not respect the geometry of the manifold sufficiently, e.g. mapping the earth to a plane causes discontinuities at the boundaries. This can be overcome by learning distributions on overlapping charts that together span the full manifold [Kalatzis et al., 2021]. An orthogonal solution is to embed the data and add noise to it in the off-manifold directions, so that the distribution can be learnt directly in an embedding space $\mathbb{R}^m$ [Brofos et al., 2021]; this only gives access to an ELBO instead of the exact density. Our method also works in the embedding space so that it respects the geometry of the manifold, but directly optimizes the likelihood on the manifold.

**Specialized architectures** take advantage of the specific geometry of a certain kind of manifold to come up with special coupling blocks for building normalizing flows such as $SO(3)$ [Liu et al., 2023], $SU(d), U(d)$ [Boyda et al., 2021, Kanwar et al., 2020]; hyperbolic space [Bose et al., 2020]; tori and spheres [Rezende et al., 2020]. Manifold free-form flows are not restricted to one particular manifold, but can be easily applied to any manifold for which an embedding and a projection to the manifold is known. As such, our model is an alternative to all of the above specialized architectures.

**Continuous time models** build a generative model based on parameterizing an ODE or SDE on any Riemannian manifold, meaning that they specify the (stochastic) differential equation in the tangent space [Rozen et al., 2021, Falorsi, 2021, Falorsi and Forré, 2020, Huang et al., 2022, Mathieu and Nickel, 2020, De Bortoli et al., 2022, Chen and Lipman, 2024, Lou et al., 2020, Ben-Hamu et al., 2022]. These methods come with the disadvantage that sampling and density evaluation integrates the ODE or SDE, requiring many function evaluations. Our manifold free-form flows do not require repeatedly evaluating the model, a single function call followed by a projection is sufficient.

At its core, our method generalizes the recently introduced free-form flow (FFF) framework [Draxler et al., 2024] based on an estimator for the gradient of the change of variables formula [Sorrenson et al., 2024]. We give more details in Section 3.1.

# 3 Background

In this section, we provide the background for our method: We present an introduction to free-form flows and Riemannian manifolds.

## 3.1 Free-form flows

Free-form flows are a class of generative models that generalize normalizing flows to work with arbitrary feed-forward neural network architectures [Draxler et al., 2024].

**Euclidean change-of-variables** Traditionally, normalizing flows are based on invertible neural networks (INNs, see Kobyzev et al. [2021] for an overview) that learn an invertible transformation $z = f_\theta(x)$ mapping from data $x \in \mathbb{R}^n$ to latent codes $z \in \mathbb{R}^n$. This gives an explicitly parameterized probability density $p_\theta(x)$ via the change-of-variables:

$$\log p_\theta(x) = \log p_Z(f_\theta(x)) + \log |f'_\theta(x)|, \tag{1}$$

where $f'_\theta(x) \in \mathbb{R}^{n \times n}$ is the Jacobian matrix of $f_\theta(x)$ with respect to $x$, evaluated at $x$; $|f'_\theta(x)|$ is its absolute determinant. The distribution of latent codes $p_Z(z)$ is chosen such that the log-density is easy to evaluate and it is easy to sample from, such as a standard normal. Normalizing flows can be trained by minimizing the negative log-likelihood over the training data distribution:

$$\min_\theta \mathcal{L}_{\text{NLL}} = \min_\theta \mathbb{E}_{p_{\text{data}}(x)}[-\log p_\theta(x)]. \tag{2}$$

This is equivalent to minimizing the Kullback-Leibler-divergence between the true data distribution and the parameterized distribution $\text{KL}(p_{\text{data}} \| p_\theta)$. Sampling from the model is achieved by pushing samples from the latent distribution $z \sim p_Z$ through the inverse of the learned function: $x = f_\theta^{-1}(z) \sim p_\theta$.

**Euclidean gradient estimator** Naively computing the volume change $\log |f'_\theta(x)|$ in Eq. (1) is expensive since it contains the full Jacobian matrix $f'_\theta(x) \in \mathbb{R}^{n \times n}$, requiring $\mathcal{O}(n)$ automatic differentiation steps to compute. Normalizing flow architectures usually avoid this expensive computation by further restricting the architecture to allow fast computation. Luckily, even if such a shortcut is not available, its *gradient*, which is all we need for training, can still be efficiently estimated as follows:

**Theorem 1** (Volume change gradient estimator, Draxler et al. [2024]). *Let $f_\theta : \mathbb{R}^n \to \mathbb{R}^n$ be a diffeomorphism. Let $v \in \mathbb{R}^n$ be a random variable with zero mean and unit covariance. Then, the derivative of the volume change has the following trace expression, where $z = f_\theta(x)$:*

$$\nabla_\theta \log |f_\theta'(x)| = \text{tr}((\nabla_\theta f_\theta'(x))f_\theta'^{-1}(z)) \tag{3}$$

$$= \mathbb{E}_v \left[ v^T (\nabla_\theta f_\theta'(x)) f_\theta^{-1'}(z) v \right]. \tag{4}$$

Replacing the expected value over $v$ by a single sample, and using a stop-grad (`SG`) operation, Theorem 1 allows us to compute a term whose gradient is an unbiased estimator for the gradient of Eq. (2):

$$\nabla_\theta \mathcal{L}_{\text{NLL}}(x) \approx \nabla_\theta (-\log p_Z(z) - v^T f_\theta'(x) \texttt{SG}(f_\theta^{-1'}(z)v)). \tag{5}$$

Comparing Eqs. (1) and (5) reveals that $\log |f_\theta'(x)|$ is replaced by a single vector-Jacobian $v^T f_\theta'(x)$ and a Jacobian-vector product $f_\theta^{-1'}(z)v$, each of which require only one automatic differentiation operation. Note that while the gradient estimate is unbiased, computing the term in the brackets is not informative about $\mathcal{L}_{\text{NLL}}$. Thus, for density estimation at inference, Eq. (1) is explicitly evaluated using the full Jacobian.

**Free-form architectures**  The central idea of free-form flows is to soften the restriction that the learned model be invertible. Instead, they learn two separate networks, an encoder $f_\theta$ and a decoder $g_\phi$ coupled by a reconstruction loss, circumventing the need for an invertible neural network $f_\theta$:

$$\mathcal{L}_{\text{R}} = \mathbb{E}_{p_{\text{data}}(x)} \left[ \|g_\phi(f_\theta(x)) - x\|^2 \right]. \tag{6}$$

Together, this gives the loss of free-form flows with $\beta$, the reconstruction weight as a hyperparameter:

$$\mathcal{L}_{\text{FFF}} = \mathcal{L}_{\text{NLL}} + \beta \mathcal{L}_{\text{R}}. \tag{7}$$

This allows replacing constrained invertible architectures with free-form neural networks. Since $f_\theta$ is not restricted to efficiently compute the volume change, free-form flows use Eq. (5) to compute the gradient of $\mathcal{L}_{\text{NLL}}$. To compute Eq. (5), free-form flows approximate $f_\theta^{-1'}(z)$ (which is not tractable) by $g_\phi'(z)$ during training:

$$\nabla_\theta \mathcal{L}_{\text{NLL}}(x) \approx \nabla_\theta (-\log p_Z(z) - v^T f_\theta'(x) \texttt{SG}(g_\phi'(z)v)). \tag{8}$$

For density estimation at inference, Draxler et al. [2024] recommend using the explicit decoder Jacobian for the volume change.

## 3.2 Riemannian manifolds

A manifold is a fundamental concept in mathematics, providing a framework for describing and analyzing spaces that locally resemble Euclidean space, but may have different global structure. For example, a small region on a sphere is similar to the Euclidean plane, but walking in a straight line on the sphere in any direction will return back to the starting point, unlike on a plane.

Mathematically, an $n$-dimensional manifold, denoted as $\mathcal{M}$, is a space where every point has a neighborhood that is topologically equivalent to $\mathbb{R}^n$. A Riemannian manifold $(\mathcal{M}, G)$ extends the concept of a manifold by adding a Riemannian metric $G$ which introduces a notion of distances and angles. At each point $x$ on the manifold, there is an associated tangent space $\mathcal{T}_x\mathcal{M}$ which is an $n$-dimensional Euclidean space, characterizing the directions in which you can travel and still stay on the manifold. The metric $G$ acts in this space, defining an inner product between vectors. From this inner product, we can compute the length of paths along the manifold, distances between points as well as volumes (see next section).

In this paper, we consider Riemannian manifolds globally embedded into an $m$-dimensional Euclidean space $\mathbb{R}^m$, with $n \leq m$. Embedding means that we represent a point on the manifold $x \in \mathcal{M}$ as a vector in $\mathbb{R}^m$ confined to an $n$-dimensional subspace; we write $x \in \mathcal{M} \subseteq \mathbb{R}^m$ and denote by $\pi : \mathbb{R}^m \to \mathcal{M}$ a projection from the embedding space to the manifold. A global embedding is a smooth, injective mapping of the entire manifold into $\mathbb{R}^m$, its smoothness preserving the topology.

In most cases, we work with, but are not limited to, isometrically embedded manifolds, meaning that the metric is inherited from the ambient Euclidean space. Intuitively, this means that the length of a

Table 2: Manifolds, a global embedding and corresponding projections considered in this paper.

| Manifold | Dimension $n$ | Embedding | Projection |
|---|---|---|---|
| Generic | $\mathrm{rank}(\pi'(\pi(x)))$ | $\{x \in \mathbb{R}^m : \pi(x) = x\}$ | $x \mapsto \pi(x)$ |
| Rotations $SO(d)$ | $(d-1)d/2$ | $\{Q \in \mathbb{R}^{d \times d} : QQ^T = I, \det Q = 1\}$ | $R \mapsto \arg\min_{Q \in SO(d)} \|Q - R\|_F$; see Eq. (102) |
| Sphere $\mathbb{S}^n$ | $n$ | $\{x \in \mathbb{R}^{n+1} : \|x\| = 1\}$ | $x \mapsto x/\|x\|$ |
| Torus $\mathbb{T}^n = (\mathbb{S}^1)^n$ | $n$ | $\{X \in \mathbb{R}^{n \times 2} : \|X_i\| = 1 \text{ for } i = 1...n\}$ | $X_i \mapsto X_i/\|X_i\|$ for $i = 1...n$ |
| Hyperbolic $\mathbb{H}^n$ | $n$ | $\{x \in \mathbb{R}^n : \|x\| < 1\}$ | $x \mapsto x \min\{1, (1-\epsilon)/\|x\|\}$ |

path on the manifold is just the length of the path in the embedding space. We note that due to the Nash embedding theorem [Nash, 1956], every Riemannian manifold has a smooth isometric embedding into Euclidean space of some finite dimension, so in this sense using isometric embeddings is not a limitation. Nevertheless, for some manifolds (especially with negative curvature, e.g. hyperbolic space) there may not be a sensible isometric embedding.

## 4  Manifold free-form flows

The free-form flow (FFF) framework allows training any pair of parameterized encoder $f_\theta(x)$ and decoder $g_\phi(z)$ as a generative model, see Section 3.1. In this section, we demonstrate how to generalize the steps in Section 3.1 to arbitrary Riemannian manifolds. Note that for simplicity, we choose the same manifold in data and latent spaces, i.e. $\mathcal{M}_X = \mathcal{M}_Z = \mathcal{M}$, but the method readily applies to $\mathcal{M}_X \neq \mathcal{M}_Z$ or $G_X \neq G_Z$ as long as they are topologically compatible, like a sphere and a closed 3D surface without holes. The detailed derivations in the appendix consider this generalization.

**Manifold change of variables**   The volume change on manifolds generalizes the Euclidean variant in Eq. (1) by (a) considering the change of volume in the tangent space and (b) accounting for volume change due to changes in the metric:

**Theorem 2** (Manifold change of variables). *Let $(\mathcal{M}, G)$ be a $n$-dimensional Riemannian manifold embedded in $\mathbb{R}^m$, i.e., $\mathcal{M} \subseteq \mathbb{R}^m$. Let $p_X$ be a probability distribution on $\mathcal{M}$ and let $f : \mathcal{M} \to \mathcal{M}$ be a diffeomorphism. Let $p_Z$ be the pushforward of $p_X$ under $f$ (i.e., if $p_X$ is the probability density of $X$, then $p_Z$ is the probability density of $f(X)$).*

*Let $x \in \mathcal{M}$. Define $Q \in \mathbb{R}^{m \times n}$ as an orthonormal basis for $\mathcal{T}_x \mathcal{M}$ and $R \in \mathbb{R}^{m \times n}$ as an orthonormal basis for $\mathcal{T}_{f(x)} \mathcal{M}$.*

*Then, the probability densities $p_X$ and $p_Z$ are related under the change of variables $x \mapsto f(x)$ by the following equation:*

$$\log p_X(x) = \log p_Z(f(x)) + \log |R^T f'(x) Q| + \tfrac{1}{2} \log \frac{|R^T G(f(x)) R|}{|Q^T G(x) Q|}. \tag{9}$$

*where $Q$ and $R$ depend on $x$ and $f(x)$, respectively, although this dependency is omitted for brevity.*

To give an intuition for this result, Fig. 2 shows how the volume change is computed for an isometrically embedded manifold, that is $G = I$ so that $|R^T G R| = |Q^T G Q| = 1$. This simplifies Eq. (9) to:

$$\log p_X(x) = \log p_Z(f(x)) + \log |R^T f'(x) Q|. \tag{10}$$

This is very similar to the familiar change of variables formula in the Euclidean case in Eq. (1), the only difference being that the determinant is evaluated on the $n \times n$ projection of $f'(x)$ into the tangent spaces. These projections are necessary as the Jacobian of $f$ is singular in the embedding space, since its action is restricted to the local tangent spaces. See the full proof in Appendix A.1.

**Manifold gradient estimator**   We now generalize the volume change gradient estimator in Theorem 1 to an invertible function on the manifold $f_\theta : \mathcal{M} \to \mathcal{M}$. We find that taking the gradient of the manifold change of variables in Eq. (9) results in essentially the same computation as in the Euclidean case, but the trace in is now evaluated in the local tangent space:

**Theorem 3.** *Under the assumptions of Theorem 2 with $f = f_\theta$. Let $v \in \mathbb{R}^m$ be a random variable with zero mean and covariance $RR^T$. Then, the derivative of the change of variables term has the*

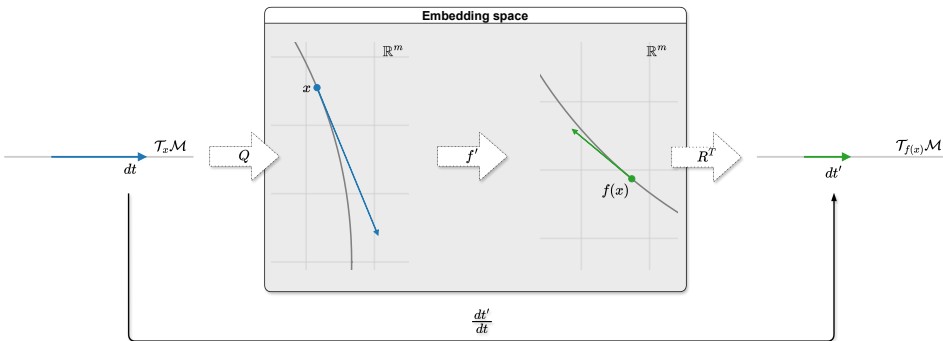

Figure 2: Computation of the volume change in the tangent space of the manifold: The manifold change of variables formula in Eq. (10) requires to compute the change of a volume element in the tangent spaces under f, which in this example is given by the ratio of lengths of $dt$ and $dt'$. Since $f$ is a map in the embedding space, $f'(x)$ defines a linear map between vectors from the embedding space. To correctly compute the change in volume, we use $Q$ and $R$ to change coordinates to the intrinsic tangent spaces, resulting in the linear map $R^T f'(x)Q : \mathcal{T}_x\mathcal{M} \to \mathcal{T}_{f(x)}\mathcal{M}$, which maps $dt$ to $dt'$.

*following trace expression, where $z = f_\theta(x)$:*

$$\nabla_\theta \log |R^T f'_\theta(x)Q| = \text{tr}(R^T(\nabla_\theta f'_\theta(x))f_\theta^{-1'}(z)R) \tag{11}$$

$$= \mathbb{E}_v \left[ v^T(\nabla_\theta f'_\theta(x))f_\theta^{-1'}(z)v \right]. \tag{12}$$

This shows that the adaptation of free-form flows for an invertible function $f$ to isometrically embedded manifolds is remarkably simple (see full proof in Appendix A.2; if the manifold is not isometrically embedded, add the corresponding term in Eq. (9)):

$$\nabla_\theta \mathcal{L}_{\mathcal{M}\text{-NLL}} = \nabla_\theta \mathbb{E}_{x,v} \left[ -\log p_Z(z) - v^T f'_\theta(x)\texttt{SG}[f_\theta^{-1'}(z)v] \right]. \tag{13}$$

The only change is that $v$ must have covariance $RR^T$ rather than the identity. We achieve this by sampling standard normal vectors $\tilde{v} \in \mathbb{R}^m$ and then projecting them into the tangent space using the Jacobian of the projection function:

$$v = \pi'(f_\theta(x))\tilde{v}. \tag{14}$$

Constructing $v$ like this fulfills the conditions of Theorem 3 because $\mathbb{E}_v[v] = 0$, and:

$$\text{Cov}[v] = \mathbb{E}_{\tilde{v}} \left[ \pi'(f_\theta(x))\tilde{v}\tilde{v}^T\pi'(f_\theta(x))^T \right] = \pi'(f_\theta(x))\pi'(f_\theta(x))^T = RR^T. \tag{15}$$

Just like [Sorrenson et al., 2024, Draxler et al., 2024], we further normalize $v$ to reduce the variance of the trace estimator. Equation (13) now allows training invertible architectures on manifolds even if the volume change $\log |R^T f'_\theta(x)Q|$ is not tractable.

Despite using a stochastic estimator for the gradient, we argue in Appendix A.5 that the scaling of the estimator variance with dimension is comparable to the variance due to stochasticity in flow matching and similar methods.

**Free-form manifold-to-manifold neural networks** As discussed in Section 2, invertible architectures have to be specially constructed for each manifold. To overcome this limitation, we now soften the constraint that the learned model be analytically invertible. Instead, we learn a pair of free-form manifold-to-manifold neural networks, an encoder $f_\theta(x)$ and a decoder $g_\phi(z)$ as arbitrary functions on the manifold:

$$f_\theta(x) : \mathcal{M} \to \mathcal{M}, \quad g_\phi(z) : \mathcal{M} \to \mathcal{M}. \tag{16}$$

We choose to fulfill Eq. (16) using feed-forward neural networks $\tilde{f}_\theta, \tilde{g}_\phi : \mathbb{R}^m \to \mathbb{R}^m$ working in an embedding space $\mathbb{R}^m$ of $\mathcal{M}$, but ensure that their outputs lie on the manifold by appending a projection $\pi : \mathbb{R}^m \to \mathcal{M}$, mapping points from the embedding space $\mathbb{R}^m$ to the manifold $\mathcal{M}$:

$$f_\theta(x) = \pi(\tilde{f}_\theta(x)), \quad g_\phi(z) = \pi(\tilde{g}_\phi(z)). \tag{17}$$

Figure 1 illustrates this for an example on a circle $\mathcal{M} = \mathbb{S}^1$.

Just like in the Euclidean case, we employ a reconstruction loss to make $f_\theta$ and $g_\phi$ approximately inverse to one another:

$$\mathcal{L}_{\mathrm{R}} = \mathbb{E}_{p_{\mathrm{data}}}[\|g_\phi(f_\theta(x)) - x\|^2]. \tag{18}$$

We measure the distance in the embedding space; one can modify this to use an on-manifold distance (e.g. great circle distance for the sphere) but we find that ambient Euclidean distance works well in practice, since it is almost identical for small distances and this is the regime we work in.

This allows us to substitute $f_\theta^{-1\prime}(z) \approx g_\phi'(z)$ in Eq. (13):

$$\nabla_\theta \mathcal{L}_{\mathcal{M}\text{-NLL}} \approx \nabla_\theta \mathbb{E}_{x,v} \left[ -\log p_Z(z) - v^T f_\theta'(x) \mathtt{SG}[g_\phi'(z)v] \right]. \tag{19}$$

In Theorem 5 we show that the error of the gradient estimator is bounded by a measure of the mismatch between the encoder and decoder Jacobian matrices. When the encoder and decoder are true inverses, the error reaches zero.

**Regularization and final loss** We find that adding the following two regularizations to the loss improve the stability and performance of our models. Firstly, the reconstruction loss on points sampled uniformly from the data manifold:

$$\mathcal{L}_{\mathrm{U}} = \mathbb{E}_{x \sim \mathcal{U}(\mathcal{M})}[\|g_\phi(f_\theta(x)) - x\|^2], \tag{20}$$

helps ensure that we have a globally consistent mapping between the data and latent manifolds in low data regions. Secondly, the squared distance between the output of $\tilde{f}_\theta$ and its projection to the manifold:

$$\mathcal{L}_{\mathrm{P}} = \mathbb{E}_{p_{\mathrm{data}}(x)}[\|\tilde{f}_\theta(x) - f_\theta(x)\|^2] \tag{21}$$

discourages the output of $\tilde{f}_\theta$ from entering unprojectable regions, for example the origin when the manifold is $\mathbb{S}^n$. The same regularizations can be applied starting from the latent space.

The full loss is:

$$\mathcal{L} = \mathcal{L}_{\mathcal{M}-\mathrm{NLL}} + \beta_{\mathrm{R}} \mathcal{L}_{\mathrm{R}} + \beta_{\mathrm{U}} \mathcal{L}_{\mathrm{U}} + \beta_{\mathrm{P}} \mathcal{L}_{\mathrm{P}} \tag{22}$$

where the gradient of $\mathcal{L}_{\mathcal{M}-\mathrm{NLL}}$ is computed using Eq. (19), and $\beta_{\mathrm{R}}$, $\beta_{\mathrm{U}}$ and $\beta_{\mathrm{P}}$ are hyperparameters. We give our choices in Appendix B.

## 5 Experiments

We now demonstrate the practical performance of manifold free-form flows on various manifolds. We choose established experiments to ensure comparability with previous methods, and find:

- M-FFF matches or outperforms previous single-step methods. M-FFF uses a simple ResNet architecture, whereas previous methods were specialized to the given manifolds, hindering adoption to novel manifolds.
- M-FFF generates samples faster by typically two orders of magnitude than methods sampling in several steps. Despite this great reduction in compute, it achieves a higher generative quality in several cases.

In our result tables, we mark as bold (a) the best method overall (both single- and multi-step), and (b) the best single-step method. We provide reconstruction losses of our method and all details necessary to reproduce the experiments in Appendix B. Furthermore, our code is available at `https://github.com/vislearn/FFF`. We run each experiment multiple times with different data splits and report the mean and standard deviation of those runs.

**Synthetic distribution over rotation matrices** The group of 3D rotations $SO(3)$ can be represented by rotations matrices with positive determinant, i.e., all $Q \in \mathbb{R}^{3\times3}$ with $Q^T Q = I$ and $\det Q = 1$. We choose $\mathbb{R}^{3\times3}$ as our embedding space and project to the manifold by solving the constrained Procrustes problem via SVD [Lawrence et al., 2019] (see Appendix B.2).

We evaluate M-FFF on synthetic mixture distributions proposed by De Bortoli et al. [2022] with $M$ mixture components for $M = 16, 32$ and $64$. Samples from one of the distributions and samples from our model are depicted in Fig. 3.

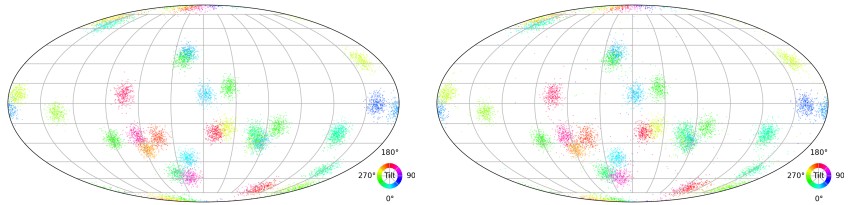

Figure 3: Manifold free-form flows on a synthetic $SO(3)$ mixture distribution with $M = 64$ mixture components proposed by De Bortoli et al. [2022]. *(Left)* 10,000 samples each from the ground truth distribution and *(right)* our model. This visualization computes three Euler angles, which fully describe a rotation matrix, and then plot the first two angles on the projection of a sphere and the last by color [Murphy et al., 2021]. We find that our model nicely samples from the distribution with few outliers between the modes.

Table 3 shows that M-FFF outperforms the normalizing flow developed for $SO(3)$ by Liu et al. [2023], as well as the diffusion-based approaches for the mixtures $M = 32$ and $64$.

Table 3: Test negative log-likelihood (NLL, $\downarrow$) on $SO(3)$ for multi-step and single-step methods. M-FFF consistently outperforms the specialized normalizing flow by Liu et al. [2023] on synthetic distributions of $SO(3)$ matrices, and outperforms multi-step methods in the cases with more mixture components. Multi-step baseline values are due to De Bortoli et al. [2022].

| | $M = 16$ | $M = 32$ | $M = 64$ | Fast inference? |
|---|---|---|---|---|
| Moser flow [Rozen et al., 2021] | $-0.85_{\pm 0.03}$ | $-0.17_{\pm 0.03}$ | $0.49_{\pm 0.02}$ | ✗: 1000 steps |
| Exp-wrapped SGM [De Bortoli et al., 2022] | $-0.87_{\pm 0.04}$ | $-0.16_{\pm 0.03}$ | $0.58_{\pm 0.04}$ | ✗: 500 steps |
| Riemannian SGM [De Bortoli et al., 2022] | $\mathbf{-0.89}_{\pm \mathbf{0.03}}$ | $-0.20_{\pm 0.03}$ | $0.49_{\pm 0.02}$ | ✗: 100 steps |
| $SO(3)$-NF [Liu et al., 2023] | $-0.81_{\pm 0.01}$ | $-0.12_{\pm 0.004}$ | $0.61_{\pm 0.01}$ | ✓ |
| M-FFF (ours) | $\mathbf{-0.87}_{\pm \mathbf{0.02}}$ | $\mathbf{-0.21}_{\pm \mathbf{0.02}}$ | $\mathbf{0.45}_{\pm \mathbf{0.02}}$ | ✓ |

**Earth data on the sphere** We evaluate manifold free-form flows on spheres with datasets from the domain of earth sciences. We use four established datasets compiled by Mathieu and Nickel [2020] for density estimation on $\mathbb{S}^2$: Volcanic eruptions [NGDC/WDS, 2022b], earthquakes [NGDC/WDS, 2022a], floods [Brakenridge, 2017] and wildfires [EOSDIS, 2020].

Figure 1 shows an example for a model trained on flood data. As the reconstruction error sometimes does not drop to a satisfactory level we employ the method described in Appendix B.1 to ensure that the measured likelihoods are accurate. Table 4 shows that M-FFF again outperforms the specialized single-step model; the performance compared to multi-step methods is mixed. We think that multi-step models have an advantage on the considered data, as there are large regions of empty space between highly concentrated data points (see density and sample plots in Appendix B.3).

Table 4: M-FFF significantly outperforms the previous single-step density estimator [Peel et al., 2001] on the sphere on real-world earth datasets in terms of negative log-likelihood (lower is better). Baseline values are collected from De Bortoli et al. [2022], Huang et al. [2022], Chen and Lipman [2024].

| | Volcano | Earthquake | Flood | Fire | Fast inference? |
|---|---|---|---|---|---|
| Riemannian CNF [Mathieu and Nickel, 2020] | $-6.05_{\pm 0.61}$ | $0.14_{\pm 0.23}$ | $1.11_{\pm 0.19}$ | $-0.80_{\pm 0.54}$ | ✗: $\sim$100 steps |
| Moser flow [Rozen et al., 2021] | $-4.21_{\pm 0.17}$ | $-0.16_{\pm 0.06}$ | $0.57_{\pm 0.10}$ | $-1.28_{\pm 0.05}$ | ✗: $\sim$100 steps |
| Stereographic score-based [De Bortoli et al., 2022] | $-3.80_{\pm 0.27}$ | $-0.19_{\pm 0.05}$ | $0.59_{\pm 0.07}$ | $-1.28_{\pm 0.12}$ | ✗: $\sim$100 steps |
| Riemannian score-based [De Bortoli et al., 2022] | $-4.92_{\pm 0.25}$ | $-0.19_{\pm 0.07}$ | $0.45_{\pm 0.17}$ | $-1.33_{\pm 0.06}$ | ✗: $\sim$100 steps |
| Riemannian diffusion [Huang et al., 2022] | $-6.61_{\pm 0.97}$ | $\mathbf{-0.40}_{\pm \mathbf{0.05}}$ | $0.43_{\pm 0.07}$ | $-1.38_{\pm 0.05}$ | ✗: $>$100 steps |
| Riemannian flow matching [Chen and Lipman, 2024] | $\mathbf{-7.93}_{\pm \mathbf{1.67}}$ | $-0.28_{\pm 0.08}$ | $\mathbf{0.42}_{\pm \mathbf{0.05}}$ | $\mathbf{-1.86}_{\pm \mathbf{0.11}}$ | ✗: 1000 steps |
| Mixture of Kent [Peel et al., 2001] | $-0.80_{\pm 0.47}$ | $0.33_{\pm 0.05}$ | $0.73_{\pm 0.07}$ | $-1.18_{\pm 0.06}$ | ✓ |
| M-FFF (ours) | $\mathbf{-2.25}_{\pm \mathbf{0.02}}$ | $\mathbf{-0.23}_{\pm \mathbf{0.01}}$ | $\mathbf{0.51}_{\pm \mathbf{0.01}}$ | $\mathbf{-1.19}_{\pm \mathbf{0.03}}$ | ✓ |
| Datset size | 827 | 6120 | 4875 | 12809 | |

Table 5: M-FFF consistently outperforms normalizing flows specialized to tori [Rezende et al., 2020] on torus datasets, without requiring the development of a specialized architecture. In addition, our method comes close to the performance of the multi-step methods and even outperforms them on the Glycine dataset. Baseline values are due to Huang et al. [2022], Chen and Lipman [2024].

| | General | Glycine | Proline | Pre-Pro | RNA | Fast inference? |
|---|---|---|---|---|---|---|
| Riemannian diffusion [Huang et al., 2022] | $1.04_{\pm 0.012}$ | $1.97_{\pm 0.012}$ | $\mathbf{0.12}_{\pm 0.011}$ | $1.24_{\pm 0.004}$ | $-3.70_{\pm 0.592}$ | ✗: ~1000 steps |
| Riemannian flow matching [Chen and Lipman, 2024] | $\mathbf{1.01}_{\pm 0.025}$ | $1.90_{\pm 0.055}$ | $0.15_{\pm 0.027}$ | $\mathbf{1.18}_{\pm 0.055}$ | $\mathbf{-5.20}_{\pm 0.067}$ | ✗: 1000 steps |
| Mixture of power spherical [Huang et al., 2022] | $1.15_{\pm 0.002}$ | $2.08_{\pm 0.009}$ | $0.27_{\pm 0.008}$ | $1.34_{\pm 0.019}$ | $4.08_{\pm 0.368}$ | ✓ |
| Circular Spline Coupling Flows [Rezende et al., 2020] | $\mathbf{1.03}_{\pm 0.01}$ | $1.91_{\pm 0.04}$ | $0.21_{\pm 0.08}$ | $1.24_{\pm 0.04}$ | $-4.01_{\pm 0.24}$ | ✓ |
| M-FFF (ours) | $\mathbf{1.03}_{\pm 0.02}$ | $\mathbf{1.89}_{\pm 0.05}$ | $\mathbf{0.17}_{\pm 0.08}$ | $\mathbf{1.23}_{\pm 0.04}$ | $\mathbf{-4.27}_{\pm 0.09}$ | ✓ |

Table 6: Test NLL on Stanford bunny data proposed by [Chen and Lipman, 2024], living on a manifold with nontrivial curvature (see Fig. 1). M-FFF outperforms the multi-step model for datasets with more modes.

| | $k = 10$ | $k = 50$ | $k = 100$ | Fast inference? |
|---|---|---|---|---|
| Riemannian Flow Matching (w/ diffusion) [Chen and Lipman, 2024] | $1.16_{\pm 0.02}$ | $1.48_{\pm 0.01}$ | $1.53_{\pm 0.01}$ | ✗: 1000 steps |
| Riemannian Flow Matching (w/ biharmonic) [Chen and Lipman, 2024] | $\mathbf{1.06}_{\pm 0.05}$ | $1.55_{\pm 0.01}$ | $1.49_{\pm 0.01}$ | ✗: 1000 steps |
| M-FFF (ours) | $\mathbf{1.21}_{\pm 0.01}$ | $\mathbf{1.34}_{\pm 0.01}$ | $\mathbf{1.28}_{\pm 0.01}$ | ✓ |

**Torsion angles of molecules on tori**  To benchmark manifold free-form flows on tori $\mathbb{T}^n$, we follow [Huang et al., 2022] and evaluate our model on two datasets from structural biology. We consider the torsion (dihedral) angles of the backbone of protein and RNA substructures respectively.

We represent a tuple of angles $(\phi_1, \ldots, \phi_n) \in [0, 2\pi]^n$ by mapping each angle to a position on a circle: $X_i = (\cos \phi_i, \sin \phi_i) \in \mathbb{S}^1$. Then we stack all $X_i$ into a matrix $X \in \mathbb{R}^{n \times 2}$, compare Table 2.

The first dataset is comprised of 500 proteins assembled by [Lovell et al., 2003] and is located on $\mathbb{T}^2$. The three dimensional arrangement of a protein backbone can be described by the so called Ramachandran angles [Ramachandran et al., 1963] $\Phi$ and $\Psi$, which represent the torsion of the protein backbone around the $N$-$C_\alpha$ and $C_\alpha$-$C$ bonds. The data is split into four distinct subsets *General*, *Glycine*, *Proline* and *Pre-Proline*, depending on the residue of each substructure.

The second dataset is extracted from a subset of RNA structures introduced by Murray et al. [2003]. As the RNA backbone structure can be characterized by seven torsion angles, in this case we are dealing with data on $\mathbb{T}^7$.

We report negative log-likelihoods in Table 5, finding that M-FFF outperforms a circular spline coupling flow, a normalizing flow particularly developed for data on tori [Rezende et al., 2020] as well as the multi-step methods on one of the datasets. In addition to the quantitative results, we show the log densities of the M-FFF models for the four protein datasets inFig. 5 in Appendix B.4 .

**Toy distributions on hyperbolic space**  We apply M-FFF to the Poincaré ball model, which embeds the 2-dimensional hyperbolic space $\mathbb{H}^2$ of constant negative curvature -1 in the 2-dimensional Euclidean space $\mathbb{R}^2$, as specified in Table 2. As this embedding is not isometric, and distances between points grow when moving away from the origin, we must include the last term of Eq. (9) when changing variables under a map on this embedded manifold.

We show that M-FFF can be applied to non-isometric embeddings using Eq. (9) and visualize learned densities in Fig. 1 and in Fig. 6 in Appendix B.5 for several toy datasets defined on the 2-dimensional Poincaré ball model. Further details can be found in Appendix B.5.

**Manifold with non-trivial curvature**  Finally, we follow Chen and Lipman [2024] and train M-FFF given by synthetic distributions on the Stanford bunny [Turk and Levoy, 1994] on the data provided with their paper, see Fig. 1. The natural embedding of this mesh is $\mathbb{R}^3$, and we train a separate neural network to project from the embedding space to the mesh. This ensures that the projection is continuously differentiable, which we identify to be important for stable gradients.

Table 6 shows that M-FFF performs well on this manifold, outperforming Riemannian flow matching in two out of three cases. This experiment underlines the flexibility of our model: We only need a projection function to the manifold in order to train a generative model.

# 6    Limitations

Manifold Free-From Flows achieve high generative quality on manifolds despite the approximations made during training: First, the exact inverse of the encoder Jacobian is approximated by the decoder Jacobian, which is implicitly regularized via the reconstruction loss (see Eq. (19)). Second, the final gradient computation in Eq. (8) is estimated with a single $v$ for each item in the batch, adding noise to the system.

At inference time, the negative log-likelihoods we report in all tables are based on the *decoder* Jacobian. We choose this because even if the decoder ends up not to be invertible after training (that is several latent codes $z$ yield the same generation $x = g_\phi(z)$), the computed densities are a conservative estimate of the true probability density. The downside is that if the reconstruction loss is high, the likelihoods become inaccurate, see Appendix B.1 for details. We therefore ensure that the final reconstruction losses are vanishing in Table 8.

From a high level perspective, we observe that M-FFF performs less favorable compared to multi-step methods when the density changes sharply or very low density regions are present.

# 7    Conclusion

In this paper, we present Manifold Free-Form Flows (M-FFF), a generative model designed for manifold data. To the best of our knowledge, it is the first generative model on manifolds with single-step sampling and density estimation readily applicable to arbitrary Riemannian manifolds. This significantly accelerates inference and allows for deployment on edge devices.

M-FFF matches or outperforms single-step architectures specialized to particular manifolds. It also surpasses multi-step methods in several cases, despite reducing the inference compute by typically two orders of magnitude.

Adapting M-FFF to new manifolds is straightforward and only requires selecting an embedding space and a projection to the manifold. In contrast, competing multi-step methods are more challenging to adapt as they require implementing a diffusion process or computing distances on the manifold.

## Acknowledgements

This work is supported by Deutsche Forschungsgemeinschaft (DFG, German Research Foundation) under Germany's Excellence Strategy EXC-2181/1 - 390900948 (the Heidelberg STRUCTURES Cluster of Excellence). It is also supported by the Vector Stiftung in the project TRINN (P2019-0092), and by Informatics for Life funded by the Klaus Tschira Foundation. AR acknowledges funding from the Carl-Zeiss-Stiftung. The authors acknowledge support by the state of Baden-Württemberg through bwHPC and the German Research Foundation (DFG) through grant INST 35/1597-1 FUGG.

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

# A   Free-form flows on Riemannian manifolds

In this appendix, we will focus on intuitive definitions of concepts from topology and differential geometry. For a more rigorous treatment of these concepts, see [Jost, 2008].

An $n$-dimensional manifold $\mathcal{M}$ is a space where every point $x$ has a neighborhood which is homeomorphic to an open subset of $\mathbb{R}^n$. Intuitively, this means that there is a small region of $\mathcal{M}$ containing $x$ which can be bent and stretched in a continuous way to map onto a small region in $\mathbb{R}^n$. This is what is meant when we say that the manifold locally resembles $\mathbb{R}^n$. If all these maps from $\mathcal{M}$ to $\mathbb{R}^n$ are also differentiable then the manifold itself is differentiable, as long as there is a way to connect up the local neighborhoods in a differentiable and consistent way.

The tangent space of the manifold at $x$, denoted $\mathcal{T}_x\mathcal{M}$, is an $n$-dimensional Euclidean space, which is a linearization of the manifold at $x$: if we zoom in to a very small region around $x$ the manifold looks flat, and this flat Euclidean space is aligned with the tangent space. Because the tangent space is a linearization of the manifold, this is where derivatives on the manifold live, e.g. if $f : \mathcal{M}_X \to \mathcal{M}_Z$ is a map between two manifolds, then the Jacobian $f'(x)$ is a linear map from $\mathcal{T}_x\mathcal{M}_X$ to $\mathcal{T}_{f(x)}\mathcal{M}_Z$.

A Riemannian manifold $(\mathcal{M}, G)$ is a differentiable manifold which is equipped with a Riemannian metric $G : \mathcal{T}_x\mathcal{M} \times \mathcal{T}_x\mathcal{M} \to \mathbb{R}$ which defines an inner product on the tangent space, which allows us to calculate lengths and angles in this space. The length of a smooth curve $\gamma : [0, 1] \to \mathcal{M}$ is given by the integral of the length of its velocity vector $\gamma'(t) \in \mathcal{T}_{\gamma(t)}\mathcal{M}$. This ultimately allows us to define a notion of distance on the manifold, as the curve of minimal length connecting two points.

In the remainder of the appendix we only consider Riemannian manifolds.

## A.1   Manifold change of variables

**Embedded manifolds**   We define an $n$-dimensional manifold embedded in $\mathbb{R}^m$ via a projection function

$$\pi : \mathbb{P} \to \mathbb{R}^m \tag{23}$$

where $\mathbb{P} \subseteq \mathbb{R}^m$ is the projectable set. We require the projection to have the following properties (the first is true of all projections, the others are additional requirements):

1. $\pi \circ \pi = \pi$
2. $\pi$ is smooth on $\mathbb{P}$
3. $\mathrm{rank}(\pi'(\pi(x))) = n$ for all $x \in \mathbb{P}$

Given such a projection, we define a manifold by

$$\mathcal{M} = \{x \in \mathbb{R}^m : \pi(x) = x\} \tag{24}$$

with the tangent space

$$\mathcal{T}_x\mathcal{M} = \mathrm{col}(\pi'(x)) \tag{25}$$

where $\mathrm{col}$ denotes the column space. Since the rank of $\pi'(x)$ with $x \in \mathcal{M}$ is $n$, the tangent space is $n$-dimensional and $\mathcal{M}$ is an $n$-dimensional manifold. To avoid clutter we denote the Riemannian metric and its $m \times m$ matrix representation with $G$ interchangeably. If $\mathcal{M}$ is isometrically embedded then $G(x)$ is just the identity matrix.

The Jacobian of the projection is a projection matrix, meaning $\pi'(x)\pi'(x) = \pi'(x)$ for $x \in \mathcal{M}$. For any $v$ in the column space of $\pi'(x)$, there is a $u$ such that $v = \pi'(x)u$ and due to the projection property, $\pi'(x)v = \pi'(x)u = v$. Similarly, for any $w$ in the row space of $\pi'(x)$, $w\pi'(x) = w$. If $\pi$ is an orthogonal projection, $\pi'$ is symmetric by definition and hence the row and column spaces are identical.

**Integration on embedded manifolds**   In order to perform integration on the manifold, we cannot work directly in the $m$-dimensional coordinates of the embedding space, instead we have to introduce some local $n$-dimensional coordinates. This means that the domain of integration has to be diffeomorphic to an open set in $\mathbb{R}^n$. Since this might not be the case for the whole region of integration, we might need to partition it into such regions and perform integration on each individually (each such region, together with its map to $\mathbb{R}^n$, is known as a chart and a collection of charts is an atlas).

For example, if we want to integrate a function on the sphere, we could split the sphere into two hemispheres and integrate each separately. A hemisphere can be continuously stretched and flattened into a 2-dimensional region, whereas the whole sphere cannot without creating discontinuities.

Given an open set $U$ in $\mathbb{R}^n$, and a diffeomorphic local embedding function $\phi : U \to \mathcal{M}$, the integral of a scalar function $p : \mathcal{M} \to \mathbb{R}$ on $\phi(U) \subseteq \mathcal{M}$ is

$$\int_{\phi(U)} p dV = \int_U (p \circ \phi) \sqrt{|\phi'(u)^T G(\phi(u)) \phi'(u)|} du^1 \cdots du^n. \tag{26}$$

The integral on the right is an ordinary integral in $\mathbb{R}^n$. The quantity inside the determinant is known as the pullback metric.

**Theorem 2** (Manifold change of variables). *Let $(\mathcal{M}_X, G_X)$ and $(\mathcal{M}_Z, G_Z)$ be $n$-dimensional Riemannian manifolds embedded in $\mathbb{R}^m$, i.e., $\mathcal{M}_X, \mathcal{M}_Z \subseteq \mathbb{R}^m$, and assume they have the same global topological structure. Let $p_X$ be a probability distribution on $\mathcal{M}_X$ and let $f : \mathcal{M}_X \to \mathcal{M}_Z$ be a diffeomorphism. Let $p_Z$ be the pushforward of $p_X$ under $f$ (i.e., if $p_X$ is the probability density of $X$, then $p_Z$ is the probability density of $f(X)$).*

*Let $x \in \mathcal{M}_X$. Define $Q \in \mathbb{R}^{m \times n}$ as an orthonormal basis for $\mathcal{T}_x \mathcal{M}_X$ and $R \in \mathbb{R}^{m \times n}$ as an orthonormal basis for $\mathcal{T}_{f(x)} \mathcal{M}_Z$.*

*Then, the probability densities $p_X$ and $p_Z$ are related under the change of variables $x \mapsto f(x)$ by the following equation:*

$$\log p_X(x) = \log p_Z(f(x)) + \log |R^T f'(x) Q| + \tfrac{1}{2} \log \frac{|R^T G_Z(f(x)) R|}{|Q^T G_X(x) Q|}. \tag{27}$$

*where $Q$ and $R$ depend on $x$ and $f(x)$, respectively, although this dependency is omitted for brevity.*

Below, we provide two versions of the proof, the second being a less rigorous and more geometric variant of the first.

*Proof.* Let $\phi : \mathbb{R}^n \to \mathcal{M}_X$ be defined by $\phi(u) = \pi_X(x + Qu)$. Let $U$ be an open subset of $\mathbb{R}^n$ containing the origin which is small enough so that $\phi$ is bijective. Let $\psi : \mathbb{R}^n \to \mathcal{M}_Z$ be defined by $\psi(w) = \pi_Z(f(x) + Rw)$. Define $\varphi = \psi^{-1} \circ f \circ \phi$ and let $W = \varphi(U)$.

Note that $\phi'(u) = \pi'_X(x + Qu) \cdot Q$ and hence $\phi'(0) = \pi'_X(x) Q = Q$ (since each column of $Q$ is in $\mathcal{T}_x \mathcal{M}_X = \mathrm{col}(\pi'_X(x))$).

Similarly, $\psi'(0) = R$. Since $\psi$ is a map from $n$ to $m$ dimensions, there is not a unique function from $\mathbb{R}^m$ to $\mathbb{R}^n$ which is $\psi^{-1}$ on the manifold and there are remaining degrees of freedom in the off-manifold behavior which can result in different Jacobians. For our purposes, we define the inverse $\psi^{-1}$ such that $\psi \circ \psi^{-1}$ is an orthogonal projection onto $\mathcal{M}_Z$. This means $\psi'(\psi^{-1}(f(x)))(\psi^{-1})'(f(x)) = RR^T$ and hence $(\psi^{-1})'(f(x)) = R^T$.

Since $p_Z$ is the pushforward of of $p_X$ under $f$, the amount of probability mass contained in $\phi(U)$ is the same as that contained in $f(\phi(U)) = \psi(W)$:

$$\int_{\phi(U)} p_X(x) dV_X = \int_{\psi(W)} p_Z(z) dV_Z \tag{28}$$

and therefore:

$$\int_U p_X(\phi(u)) \sqrt{|\phi'(u)^T G_X(\phi(u)) \phi'(u)|} du^1 \cdots du^n$$
$$= \int_W p_Z(\psi(w)) \sqrt{|\psi'(w)^T G_Z(\psi(w)) \psi'(w)|} dw^1 \cdots dw^n. \tag{29}$$

Changing variables of the RHS with $w = \varphi(u)$ gives us

$$\int_U p_X(\phi(u)) \sqrt{|\phi'(u)^T G_X(\phi(u)) \phi'(u)|} du^1 \cdots du^n$$
$$= \int_U p_Z(f(\phi(u))) \sqrt{|\psi'(\varphi(u))^T G_Z(f(\phi(u))) \psi'(\varphi(u))|} \cdot \left| \frac{\partial w}{\partial u} \right| du^1 \cdots du^n. \tag{30}$$

Since $U$ was arbitrary, we can make it arbitrarily small, demonstrating that the integrands must be equal for $u = 0$:

$$p_X(x)\sqrt{|Q^T G_X(x)Q|} = p_Z(f(x))\sqrt{|R^T G_Z(f(x))R|} \cdot \left|\frac{\partial w}{\partial u}\right|. \tag{31}$$

Since $w = \psi^{-1}(f(\phi(u)))$, the Jacobian has the following form when evaluated at the origin (note $\phi(0) = x$):

$$\frac{\partial w}{\partial u} = (\psi^{-1})'(f(x)) \cdot f'(x) \cdot \phi'(0) \tag{32}$$

$$= R^T f'(x)Q. \tag{33}$$

Substituting this into the equality, rearranging and taking the logarithm gives the result:

$$\log p_X(x) = \log p_Z(f(x)) + \log |R^T f'(x)Q| + \tfrac{1}{2}\log\frac{|R^T G_Z(f(x))R|}{|Q^T G_X(x)Q|}. \tag{34}$$

$\square$

**Alternative proof** Here is a less rigorous and more geometric proof, which may be more intuitive for some readers.

*Proof.* Let $x$ be a point on $\mathcal{M}_X$. Consider a small square region $U \subseteq \mathcal{M}$ around $x$ (hypercubic region in higher dimensions). If the sides of the square are small enough, the square is approximately tangent to the manifold since the manifold looks very flat if we zoom in. Suppose $Q$ is a basis for the tangent space at $x$ and $q^1, \ldots, q^n$ are the columns of $Q$. Suppose that the sides of the square (or hypercube) are spanned by $u^i = \epsilon q^i$ for a small $\epsilon$. The volume spanned by a parallelotope (higher-dimensional analog of a parallelogram) is the square root of the determinant of the Gram matrix of inner products:

$$\mathrm{vol}(u^1, \ldots, u^n) = \sqrt{|\langle u^i, u^j\rangle|}. \tag{35}$$

The inner product is given by $G$, namely $\langle u, v\rangle = u^T G v$. We can therefore write the volume of $U$ as

$$\mathrm{vol}(U) \approx \epsilon^n\sqrt{|Q^T G Q|}. \tag{36}$$

Now consider how $U$ is transformed under $f$. It will be mapped to a region $f(U)$ on $\mathcal{M}_z$ with approximately straight edges, forming an approximate parallelotope in the tangent space at $z = f(x)$. This region will be spanned by the columns of $f'(x)\epsilon Q$ (since $f(x + u^i) \approx f(x) + f'(x)u^i$) and hence will have a volume of

$$\mathrm{vol}(f(U)) \approx \epsilon^n\sqrt{|Q^T f'(x)^T G_Z(z) f'(x)Q|} \tag{37}$$

$$= \epsilon^n\sqrt{|Q^T f'(x)^T RR^T G_Z(z) RR^T f'(x)Q|} \tag{38}$$

$$= \epsilon^n|R^T f'(x)Q|\sqrt{|R^T G_Z(z)R|} \tag{39}$$

where $R$ is a basis for the tangent space at $f(x)$. We can introduce $RR^T$ into the expression since it is a projection in the tangent space at $f(x)$ and is essentially the identity within that space. Since the RHS of $G_Z(z)$ and the LHS of $f'(x)$ both live in this tangent space, we can introduce $RR^T$ between them without changing the expression. Then in the last step we use that $|AB| = |A||B|$ for square matrices.

The probability density in $U$ and $f(U)$ should be roughly constant since both regions are very small. Since the probability mass in both regions should be the same we can write

$$p_X(x)\,\mathrm{vol}(U) \approx p_Z(f(x))\,\mathrm{vol}(f(U)) \tag{40}$$

and therefore

$$p_X(x) = p_Z(f(x))|R^T f'(x)Q|\frac{\sqrt{|R^T G_Z(f(x))R|}}{\sqrt{|Q^T G_X(x)Q|}} \tag{41}$$

where the approximation becomes exact by taking the limit of infinitesimally small $\epsilon$. Taking the logarithm, we arrive at the result of Theorem 2. $\square$

## A.2 Loss function

**Theorem 3.** *Under the assumptions of Theorem 2 with $f = f_\theta$. Let $v \in \mathbb{R}^m$ be a random variable with zero mean and covariance $RR^T$. Then, the derivative of the change of variables term has the following trace expression, where $z = f_\theta(x)$:*

$$\nabla_\theta \log |R^T f'_\theta(x)Q| = \text{tr}(R^T (\nabla_\theta f'_\theta(x)) f_\theta^{-1'}(z) R) \tag{42}$$

$$= \mathbb{E}_v \left[ v^T (\nabla_\theta f'_\theta(x)) f_\theta^{-1'}(z) v \right]. \tag{43}$$

*Proof.* For brevity, we drop the index $\theta$ and denote $g = f^{-1}$. First, a reminder that $\varphi'(u) = R^T f'(x)Q$ with $\varphi = \psi^{-1} \circ f \circ \phi$. Let $\chi = \varphi^{-1}$, i.e. $\chi = \phi^{-1} \circ g \circ \psi$. Jacobi's formula tells us that

$$\frac{d}{dt} \log |A(t)| = \text{tr}\left( \frac{dA(t)}{dt} A(t)^{-1} \right). \tag{44}$$

Note also that since $\chi(\varphi(u)) = u$, therefore $\chi'(\varphi(u))\varphi'(u) = I$ and $\chi'(\varphi(u)) = \varphi'(u)^{-1}$. Applying Jacobi's formula to $\varphi'(u)$:

$$\nabla_\theta \log |\varphi'(u)| = \text{tr}((\nabla_\theta \varphi'(u))\varphi'(u)^{-1}) \tag{45}$$

$$= \text{tr}((\nabla_\theta \varphi'(u))\chi'(\varphi(u))) \tag{46}$$

and substituting in $f$ and $g$:

$$\nabla_\theta \log |R^T f'(x)Q| = \text{tr}(\nabla_\theta (R^T f'(x)Q) Q^T g'(f(x)) R). \tag{47}$$

$Q$ does not depend on $\theta$, but $R$ depends on $f(x)$ and hence $\theta$, so it must be considered in the derivative. However,

$$\nabla_\theta \text{tr}(RR^T) = \text{tr}((\nabla_\theta R)R^T + R\nabla_\theta R^T) = 2 \text{tr}(R\nabla_\theta R^T) \tag{48}$$

and since $\text{tr}(RR^T) = \text{tr}(I)$ is a constant, $\text{tr}(R\nabla_\theta R^T) = 0$. Expanding Eq. (47):

$$\nabla_\theta \log |R^T f'(x)Q| = \text{tr}(\nabla_\theta(R^T) f'(x) Q Q^T g'(f(x)) R) + \text{tr}(R^T \nabla_\theta(f'(x)) Q Q^T g'(f(x)) R). \tag{49}$$

Since $Q$ is an orthonormal basis for $\mathcal{T}_x \mathcal{M}_X$, $QQ^T$ is a projection matrix onto $\mathcal{T}_x \mathcal{M}_X$. This is because $(QQ^T)^2 = QQ^T QQ^T = QQ^T$, using $Q^T Q = I$. As a result, $QQ^T \pi'(x) = \pi'(x)$. Since $g$ can also be written inside a projection: $g(z) = \pi_Z(g(z))$, therefore $g'(z) = \pi'_Z(\tilde{g}(z))\tilde{g}'(z)$, so $QQ^T g'(z) = g'(z)$. Note also that $f'(x)g'(f(x)) = I$ since $f \circ g = \text{id}$. This simplifies the equation:

$$\nabla_\theta \log |R^T f'(x)Q| = \text{tr}(\nabla_\theta(R^T)R) + \text{tr}(R^T \nabla_\theta(f'(x))g'(f(x)) R) \tag{50}$$

and finally

$$\nabla_\theta \log |R^T f'(x)Q| = \text{tr}(R^T \nabla_\theta(f'(x))g'(f(x)) R). \tag{51}$$

$\square$

In the above proof we used the fact that $QQ^T g'(z) = g'(z)$, where we dropped the index $\theta$ and use $g := f^{-1}$ for brevity. Can we use $RR^T f'(x) = f'(x)$ to simplify the equation further? No, we cannot, since the expression involving $f'$ is actually its derivative with respect to parameters, which may not have the same matrix structure as $f'$. Is it instead possible to use $g'(z)RR^T = g'(z)$ for simplification? If $g$ is implemented as $\pi_Z(\tilde{g}(z))$, this is not necessarily true, as $g'(z)$ might not be a map from the tangent space at $z$ to the tangent space at $g(z)$. For example, if we add a small deviation $v$ to $z$, where $v$ is orthogonal to the tangent space at $z$, then $g(z + v)$ might not equal $g(z)$. However, this would mean that derivatives in the off-manifold direction can be non-zero, meaning that $g'(z)v \neq g'(z)RR^T v = 0$ (since $RR^T$ will project $v$ to 0). We can change this by prepending $g$ by a projection:

$$g = \pi_X \circ \tilde{g} \circ \pi_Z. \tag{52}$$

If $\pi_Z$ is an orthogonal projection, meaning that $\pi'_Z$ is symmetric, the column space and row space of $\pi_Z$ will both be the same as those of $RR^T$, meaning $\pi'_Z(z)RR^T = \pi'_Z$ and hence $g'(z)RR^T = g'(z)$. This leads to the following corollary:

**Corollary 4.** *Suppose the assumptions of Theorem 2 hold with $f = f_\theta$ and the following implementation:*

$$f_\theta^{-1} = \pi_X \circ f_\theta^{-1} \circ \pi_Z \tag{53}$$

*where $\pi_Z$ is an orthogonal projection. Then the derivative of the change of variables term has the following trace expression, where $z = f_\theta(x)$:*

$$\nabla_\theta \log |R^T f_\theta'(x)Q| = \mathrm{tr}((\nabla_\theta f'(x))(f_\theta^{-1})'(z)). \tag{54}$$

*Proof.* Again, we drop the index $\theta$ and let $g = f^{-1}$ for brevity. Take the result of Theorem 3 and use the cyclic property of the trace and the properties of $g'$ discussed above:

$$\mathrm{tr}(R^T \nabla_\theta(f'(x))g'(f(x))R) = \mathrm{tr}(\nabla_\theta(f'(x))g'(f(x))RR^T) \tag{55}$$

$$= \mathrm{tr}(\nabla_\theta(f'(x))g'(f(x))). \tag{56}$$

$\square$

We use Hutchinson-style trace estimators to approximate the traces given above. This uses the property that, for a matrix $A \in \mathbb{R}^{n \times n}$ and a distribution $p(v)$ in $\mathbb{R}^n$ with unit second moment (meaning $\mathbb{E}[vv^T] = I$),

$$\mathbb{E}_{p(v)}[v^T Av] = \mathrm{tr}(\mathbb{E}_{p(v)}[v^T Av]) \tag{57}$$

$$= \mathrm{tr}(\mathbb{E}_{p(v)}[vv^T]A) \tag{58}$$

$$= \mathrm{tr}(A) \tag{59}$$

meaning that $v^T Av \approx \mathrm{tr}(A)$ is an unbiased estimate of the trace of $A$.

We have two variants of the trace estimate derived above, one evaluated in $\mathbb{R}^n$, the other in $\mathbb{R}^m$. The first can be estimated using the following equality (again dropping the index $\theta$ and using $g = f^{-1}$):

$$\mathrm{tr}(R^T \nabla_\theta(f'(x))g'(f(x))R)$$

$$= \mathbb{E}_{p(u)}[u^T R^T \nabla_\theta(f'(x))g'(f(x))Ru] \tag{60}$$

$$= \mathbb{E}_{p(v)}[v^T \nabla_\theta(f'(x))g'(f(x))v] \tag{61}$$

$$= \nabla_\theta \mathbb{E}_{p(v)}[v^T \nabla_\theta(f'(x))\mathtt{SG}[g'(f(x))]v] \tag{62}$$

where $p(u)$ has unit second moment in $\mathbb{R}^n$ and $p(v)$ is the distribution of $Ru$, which lies in the tangent space at $x$ and has unit second moment in that space by which we mean $\mathbb{E}[vv^T] = RR^T$. An example of such a distribution is the standard normal projected to the tangent space, i.e. $v = RR^T \tilde{v}$ where $\tilde{v}$ is standard normal.

In the second case, we can just sample from a distribution with unit second moment in the embedding space $\mathbb{R}^m$:

$$\mathrm{tr}(\nabla_\theta(f'(x))g'(f(x))) = \nabla_\theta \mathbb{E}_{p(v)}[v^T \nabla_\theta(f'(x))\mathtt{SG}[g'(f(x))]v]. \tag{63}$$

### A.3  Error bound

The error bound on the gradient from Draxler et al. [2024, Theorem 4.2] can be readily extended to Riemannian manifolds:

**Theorem 5.** *Carry over the assumptions of Theorem 2 with $f = f_\theta$ and let $g_\phi$ be a manifold-to-manifold function. Let $J_{f_\theta} = f_\theta'(x)$, $J_{g_\phi} = g_\phi'(z)$, and $J_{f_\theta^{-1}} = f_\theta^{-1\prime}(z)$. Then:*

$$\left| \mathrm{tr}(R^T(\nabla_\theta J_{f_\theta})J_{g_\phi}R) - \nabla_\theta \log |R^T J_{f_\theta}Q| \right| \le \|R^T(\nabla_\theta J_{f_\theta})J_{f_\theta^{-1}}R\|_F \|R^T J_{f_\theta}J_{g_\phi}R - \mathbb{I}_n\|_F. \tag{64}$$

*Proof.* The proof closely follows [Draxler et al., 2024] and utilizes the Cauchy–Schwarz inequality for the Frobenius inner product, which states that for matrices $A$ and $B$, we have $|\mathrm{tr}(A^T B)| \le \|A\|_F \|B\|_F$. Applying this to our case:

$$\left| \mathrm{tr}(R^T(\nabla_\theta J_{f_\theta})J_{g_\phi}R) - \nabla_\theta \log |R^T J_{f_\theta}Q| \right| \tag{65}$$

$$= \left| \mathrm{tr}(R^T(\nabla_\theta J_{f_\theta})J_{g_\phi}R) - \mathrm{tr}(R^T(\nabla_\theta J_{f_\theta})J_{f_\theta^{-1}}R) \right| \tag{66}$$

$$= \left| \mathrm{tr}(R^T(\nabla_\theta J_{f_\theta})(J_{g_\phi} - J_{f_\theta^{-1}})R) \right|. \tag{67}$$

We can re-express this term by introducing the identity matrix in terms of the Jacobians of $f_\theta$ and its inverse:

$$= \left| \text{tr}(R^T (\nabla_\theta J_{f_\theta}) J_{f_\theta^{-1}} R \cdot R^T (J_{f_\theta} J_{g_\phi} - \mathbb{I}_m) R) \right|. \tag{68}$$

By applying the Cauchy–Schwarz inequality, we obtain the bound:

$$\leq \|R^T (\nabla_\theta J_{f_\theta}) J_{f_\theta^{-1}} R\|_F \cdot \|R^T J_{f_\theta} J_{g_\phi} R - \mathbb{I}_n\|_F. \tag{69}$$

To further clarify, we recall the function $\varphi$ introduced in the proof of Theorem 2. This yields:

$$\mathbb{I}_n = J_{\varphi^{-1}} J_\varphi = Q^T J_{f_\theta^{-1}} R R^T J_{f_\theta} Q. \tag{70}$$

Thus, we can represent $J_{g_\phi}$ as:

$$J_{g_\phi} = Q Q^T J_{g_\phi} \tag{71}$$

$$= Q(Q^T J_{f_\theta^{-1}} R R^T J_{f_\theta} Q) Q^T J_{g_\phi} \tag{72}$$

$$= J_{f_\theta^{-1}} R R^T J_{f_\theta} J_{g_\phi}, \tag{73}$$

where we used $J_{g_\phi} = Q Q^T J_{g_\phi}$ and $J_{f_\theta^{-1}} = Q Q^T J_{f_\theta^{-1}}$ using similar reasoning to the proof of Theorem 3. $\qquad\square$

## A.4 Variance reduction

When using a Hutchinson trace estimator with standard normal $v \in R^n$, we can reduce the variance of the estimate by scaling $v$ to have length $\sqrt{n}$ (see [Girard, 1989]). The scaled variable will still have zero mean and unit covariance so the estimate remains unbiased, but the variance is reduced, with the effect especially pronounced in low dimensions.

While we can take advantage of this effect in both our options for trace estimator, the effect is more pronounced in lower dimensions, so we reduce the variance more by estimating the trace in an $n$-dimensional space rather than an $m$-dimensional space. Hence the first version of the trace estimator, where $v$ is sampled from a distribution in $\mathcal{T}_x \mathcal{M}_X$ is preferable in this regard.

Let's provide some intuition with an example. Suppose $n = 1$, $m = 2$ and $R = (1, 0)^T$. We want to estimate the trace of $A = \text{diag}(1, 0)$. Using the first estimator, we first sample $v = RR^T \tilde{v}$ with $\tilde{v}$ standard normal which results in $v = (u, 0)^T$ where $u \in \mathbb{R}$ is standard normal. Then we scale $v$ so it has length $\sqrt{n} = 1$. This results in $v = (r, 0)^T$ where $r$ is a Rademacher variable (taking the value $-1$ and $1$ with equal probability). The trace estimate is therefore $r^2 = 1$, meaning we always get the correct answer, so the variance is zero. The second estimator samples $v$ directly from a 2d standard normal, then scales it to have length $\sqrt{m} = \sqrt{2}$. Hence $v$ is sampled uniformly from the circle with radius $\sqrt{2}$. We can write $v = \sqrt{2}(\cos\theta, \sin\theta)^T$ with $\theta$ sampled uniformly in $[0, 2\pi]$. The estimate $v^T A v = 2\cos^2\theta$. This is a random variable whose mean is indeed 1 as required but has a nonzero variance, showing that the variance is higher when estimating in the $m$-dimensional space.

For this reason, we choose the first estimator, sampling $v$ in the tangent space at $x$. This also simplifies the definition of $g$, meaning that we don't have to prepend it with a projection.

## A.5 Scaling behavior

Since generating in high-dimensional spaces can raise concerns about an estimator's scaling behavior, we argue theoretically that the free-form flow estimator scales comparably to flow matching as dimensions increase. Experimental results further support this, showing that non-Riemannian free-form flows exhibit strong scaling performance in spaces up to 261 dimensions [Draxler et al., 2024].

One metric to assess scaling behavior is to consider the variance of the gradient estimator. With standard normal noise, the variance of the trace estimator $v^T A v$ is $2\|A\|_F^2$ [Hutchinson, 1989]. If we apply this to a simple linear free-form flow model $f(x) = Ax$ and $g(z) = Bz$, we find that summing up the variances of the gradient estimates with respect to each element of $A$ leads to a total variance of $2n \, \text{tr}(BB^T)$. Note that in a converged model, $BB^T$ is equal to the covariance of the data. A similar calculation for a flow matching loss of the form $\frac{1}{2}\|Ax - y\|^2$ leads to a total variance $\|x\|^2 \, \text{tr}(\Sigma)$

with $\Sigma$ the covariance of $p(y|x)$. These results are proven below. We can see that both expressions scale as $n^2$, assuming that $\|x\|^2$ and the trace terms scale as $n$. These assumptions are fulfilled if, for example, the data is Gaussian, with covariance that doesn't depend on $n$. Since the number of parameters (elements of $A$) scales as $n^2$, the variance per parameter is constant. We thus expect similar scaling behavior to flow matching, with no problems due to variance in high dimensions.

**Lemma 6.** *Let $A \in \mathbb{R}^{n \times n}$ be a matrix with entries $A_{ij}$, and let $B \in \mathbb{R}^{n \times n}$ be any matrix. Define*
$$C^{ij} = \frac{\partial A}{\partial A_{ij}} B \text{ for each } i, j. \text{ Let } v \in \mathbb{R}^n \text{ be a random vector with entries independently drawn from}$$
*the standard normal distribution. Then, the total variance of the Hutchinson estimators $v^T C^{ij} v$ for $\mathrm{tr}(C^{ij})$ over all $i, j$ is given by*

$$\text{Total Variance} = \sum_{i=1}^{n} \sum_{j=1}^{n} \mathrm{Var}\left(v^T C^{ij} v\right) = 2n\|B\|_F^2 = 2n\,\mathrm{tr}(BB^T), \tag{74}$$

*where $\|\cdot\|_F$ denotes the Frobenius norm.*

*Proof.* We know from Hutchinson [1989] that

$$\mathrm{Var}\left(v^T C^{ij} v\right) = 2\|C^{ij}\|_F^2 \tag{75}$$

Note the form of $\frac{\partial A}{\partial A_{ij}}$, using the Kronecker delta:

$$\left(\frac{\partial A}{\partial A_{ij}}\right)_{kl} = \delta_{ki}\delta_{lj}. \tag{76}$$

This implies

$$(C^{ij})_{kl} = \sum_m \left(\frac{\partial A_{km}}{\partial A_{ij}}\right) B_{ml} = \delta_{ki}\delta_{mj}B_{ml} = \delta_{ki}B_{jl}. \tag{77}$$

Thus, $C^{ij}$ has non-zero entries only in the $i$-th row, and that row is equal to the $j$-th row of $B$:

$$(C^{ij})_{kl} = \begin{cases} B_{jl} & \text{if } k = i, \\ 0 & \text{otherwise.} \end{cases} \tag{78}$$

Next, calculate the Frobenius norm of $C^{ij}$:

$$\|C^{ij}\|_F^2 = \sum_{k,l}(C^{ij})_{kl}^2 = \sum_l (B_{jl})^2 = \|B_{j:}\|_2^2, \tag{79}$$

where $B_{j:}$ denotes the $j$-th row of $B$.

Finally, sum over all $i$ and $j$ to find the total variance:

$$\text{Total Variance} = \sum_{i=1}^{n} \sum_{j=1}^{n} 2\|B_{j:}\|_2^2 = 2n \sum_{j=1}^{n} \|B_{j:}\|_2^2. \tag{80}$$

Since

$$\sum_{j=1}^{n} \|B_{j:}\|_2^2 = \|B\|_F^2, \tag{81}$$

the total variance simplifies to

$$\text{Total Variance} = 2n\|B\|_F^2. \tag{82}$$

Noting that $\|B\|_F^2 = \mathrm{tr}(BB^T)$ completes the proof. $\qquad\square$

**Lemma 7.** *Let $A \in \mathbb{R}^{n \times n}$ be a fixed matrix, $x \in \mathbb{R}^n$ a fixed vector, and let $y \in \mathbb{R}^n$ be a random vector with conditional distribution $p(y|x)$ having covariance matrix $\Sigma$. Define the loss function*

$$L(A) = \frac{1}{2}\|Ax - y\|^2.$$

*Then, the total variance of the gradient estimators $\dfrac{\partial L}{\partial A_{ij}}$ over all $i$ and $j$ under $p(y|x)$ is given by*

$$\text{Total Variance} = \|x\|^2 \, \mathrm{tr}(\Sigma).$$

*Proof.* The loss function is given by

$$L(A) = \frac{1}{2}\|Ax - y\|^2 = \frac{1}{2}(Ax - y)^T(Ax - y). \tag{83}$$

The derivative of $L(A)$ with respect to $A_{ij}$ is

$$\frac{\partial L}{\partial A_{ij}} = (Ax - y)^T \frac{\partial(Ax)}{\partial A_{ij}}. \tag{84}$$

Since $Ax$ is a vector whose $k$-th component is $(Ax)_k = \sum_{l=1}^n A_{kl}x_l$, the derivative of $(Ax)_k$ with respect to $A_{ij}$ is

$$\frac{\partial(Ax)_k}{\partial A_{ij}} = \delta_{ki}x_j, \tag{85}$$

where $\delta_{ki}$ is the Kronecker delta.

Therefore, the derivative becomes

$$\frac{\partial L}{\partial A_{ij}} = (Ax - y)_i\, x_j. \tag{86}$$

We are interested in the variance of the estimator $\dfrac{\partial L}{\partial A_{ij}}$ under $p(y|x)$. Since $A$ and $x$ are fixed, the only randomness comes from $y$. Assuming that $y$ is a random vector with mean $\mu = \mathbb{E}[y|x]$ and covariance matrix $\Sigma$, we have

$$\text{Var}\left(\frac{\partial L}{\partial A_{ij}}\right) = \text{Var}\left((Ax - y)_i\, x_j\right) = x_j^2\, \text{Var}\left((Ax - y)_i\right). \tag{87}$$

Since $(Ax - y)_i = (Ax)_i - y_i$, and $(Ax)_i$ is deterministic, it follows that

$$\text{Var}\left((Ax - y)_i\right) = \text{Var}(y_i) = \Sigma_{ii}. \tag{88}$$

Therefore,

$$\text{Var}\left(\frac{\partial L}{\partial A_{ij}}\right) = x_j^2 \Sigma_{ii}. \tag{89}$$

The total variance over all $i$ and $j$ is

$$\text{Total Variance} = \sum_{i=1}^n \sum_{j=1}^n \text{Var}\left(\frac{\partial L}{\partial A_{ij}}\right) = \sum_{i=1}^n \sum_{j=1}^n x_j^2 \Sigma_{ii} = \|x\|^2\, \text{tr}(\Sigma). \tag{90}$$

Therefore, we have proven that the total variance is $\|x\|^2\, \text{tr}(\Sigma)$. $\qquad\square$

# B  Experimental details

In accordance with the details provided in Section 4, our approach incorporates multiple regularization loss components in addition to the negative log-likelihood objective. This results in the final loss expression:

$$\mathcal{L} = \mathcal{L}_{\text{NLL}} + \beta_{\text{R}}^{x/z}\mathcal{L}_{\text{R}}^{x/z} + \beta_{\text{U}}^{x/z}\mathcal{L}_{\text{U}}^{x/z} + \beta_{\text{P}}^{x/z}\mathcal{L}_{\text{P}}^{x/z}. \tag{91}$$

For each of the terms, there is a variant in $x$- and in $z$-space, as indicated by the superscript. In detail:

The first loss $\mathcal{L}_{\text{R}}$ represents the reconstruction loss:

$$\mathcal{L}_{\text{R}}^x = \mathbb{E}_{x \sim p_{\text{data}}}[d(x, g_\phi(f_\theta(x)))], \tag{92}$$
$$\mathcal{L}_{\text{R}}^z = \mathbb{E}_{x \sim p_{\text{data}}}[d(f_\theta(x), f_\theta(g_\phi(f_\theta(x))))]. \tag{93}$$

Here $d(x, y) = \|x - y\|^2$ is the standard reconstruction loss in the embedding space. This could be replaced with a distance on the manifold. However, this would be more expensive to compute and

since we initialize networks close to identity, the distance in the embedding space is almost equal to shortest path length on the manifold.

In order to have accurate reconstructions outside of training data, we also add reconstruction losses for data uniformly sampled on the manifold, both for $x$ and $z$:

$$\mathcal{L}_{\mathrm{U}}^x = \mathbb{E}_{x \sim \mathcal{U}(\mathcal{M})}[d(x, g_\phi(f_\theta(x))], \tag{94}$$

$$\mathcal{L}_{\mathrm{U}}^z = \mathbb{E}_{x \sim \mathcal{U}(\mathcal{M})}[d(z, f_\theta(g_\phi(z))]. \tag{95}$$

Finally, we make sure that the function learned by the neural networks is easy to project by regularizing the distance between the raw outputs by the neural networks in the embedding space and the subsequent projection to the manifold (compare Eq. (17)):

$$\mathcal{L}_{\mathrm{P}}^x = \mathbb{E}_{x \sim p_{\mathrm{data}}}[\|g_\phi(f_\theta(x)) - \tilde{g}_\phi(f_\theta(x))\|^2], \tag{96}$$

$$\mathcal{L}_{\mathrm{P}}^z = \mathbb{E}_{x \sim p_{\mathrm{data}}}[\|f_\theta(x) - \tilde{f}_\theta(x)\|^2]. \tag{97}$$

If these superscripts are not specified explicitly in the following summary of experimental details, we mean $\beta^x = \beta^z$.

In all cases, we selected hyperparameter using the performance on the validation data.

## B.1 Likelihood Evaluation

Sampling from a trained model can be easily achieved by sampling from the latent distribution and performing a single pass through the decoder $g$. However, in order to evaluate the likelihood of the test set under our model, as in [Draxler et al., 2024], we need to calculate the change of variable formula w.r.t. $g^{-1}$

$$\log p_X(x) = \log p_Z(g^{-1}(x)) + \log |R^T g^{-1\prime}(x)Q| + \tfrac{1}{2}\log \frac{|R^T G_Z(g^{-1}(x))R|}{|Q^T G_X(x)Q|} \tag{98}$$

$$\approx \log p_Z(f(x)) + \log |R^T g'(f(x))^{-1}Q| + \tfrac{1}{2}\log \frac{|R^T G_Z(f(x))R|}{|Q^T G_X(x)Q|}. \tag{99}$$

Here we used the approximation $f \approx g^{-1}$. While is expensive to compute for training, it is reasonably fast to compute during inference time. To show the validity of this approximation, we report the reconstruction losses computed on the test dataset for all experiments in table 8.

**Insufficient reconstruction losses.** If the assumption of $f \approx g^{-1}$ is not sufficiently fulfilled the measured likelihoods might be inaccurate. We try to identify such cases by sampling points from $\mathcal{M}$ around the proposed latent point with small noise strength $\sigma$

$$\tilde{z} = \pi(f(x) + \mathcal{N}(0, \sigma)). \tag{100}$$

As inverse of the decoder $g$ we use the sample $\tilde{z}$ which results in the lowest reconstruction loss

$$g^{-1}(x) \approx \arg\min_{\tilde{z}} \|g(\tilde{z}) - x\|_2^2. \tag{101}$$

We also do this with samples drawn around $\tilde{x} = \pi(x + \mathcal{N}(0, \sigma))$ and uniformly from the manifold $\tilde{z} = \mathcal{U}(\mathcal{M})$. We note that in most cases except for the earth datasets the likelihoods computed in this way agree with $f \approx g^{-1}$. In case of the earth datasets, we note that the newly computed likelihoods now agree with observed model quality. Specifically, whenever reconstruction loss is low we also see agreement between the likelihoods computed via $f$ and our approximation of $g^{-1}$ via sampling. Otherwise, the disagreement between $f$ and $g^{-1}$ and the resulting drop in model quality are correctly diagnosed. Therefore, for the all $\mathbb{S}^2$ experiments we report the likelihoods computed by our approximation.

**Sampling based evaluation.** In order to evaluate our models independently of their reconstruction capabilities, we propose to also report a sample-based metric. Due to the curse of dimensionality, sample-based metrics are only tractable in low dimensions. Therefore, we report the Wasserstein-2 distance between model samples and the test dataset, including the standard deviation over multiple training runs, for all 2-dimensional manifolds in Table 7. As competing method have neither reported a sampling based evaluation metric nor published their models, we propose to use our baseline results for future benchmarking.

Table 7: Wasserstein-2 distances for all 2-dimensional manifolds.

| | |
|---|---|
| Volcano | $0.249 \; _{\pm 0.5}$ |
| Earthquakes | $0.068 \; _{\pm 0.022}$ |
| Flood | $0.047 \; _{\pm 0.010}$ |
| Fire | $0.072 \; _{\pm 0.027}$ |
| General | $0.21 \; _{\pm 0.04}$ |
| Glycine | $0.32 \; _{\pm 0.05}$ |
| Proline | $0.51 \; _{\pm 0.05}$ |
| Pre-Pro | $0.47 \; _{\pm 0.04}$ |
| Bunny ($k = 10$) | $0.09 \; _{\pm 0.02}$ |
| Bunny ($k = 50$) | $0.046 \; _{\pm 0.006}$ |
| Bunny ($k = 100$) | $0.026 \; _{\pm 0.007}$ |

Table 8: The reconstruction losses $\mathcal{L}_R$ show that the reconstructed points lie close to the original data. This justifies evaluating M-FFF via negative log-likelihoods.

| | |
|---|---|
| SO(3) ($K = 16$) | $(2.7 \pm 0.3) \times 10^{-5}$ |
| SO(3) ($K = 32$) | $(1.6 \pm 1.8) \times 10^{-4}$ |
| SO(3) ($K = 64$) | $(7.2 \pm 1.6) \times 10^{-5}$ |
| Volcano | $(5.1 \pm 0.8) \times 10^{-6}$ |
| Earthquakes | $(1.9 \pm 0.8) \times 10^{-7}$ |
| Flood | $(1.9 \pm 0.6) \times 10^{-5}$ |
| Fire | $(1.7 \pm 0.4) \times 10^{-6}$ |
| General | $(7.2 \pm 0.9) \times 10^{-6}$ |
| Glycine | $(1.6 \pm 0.2) \times 10^{-5}$ |
| Proline | $(1.3 \pm 0.3) \times 10^{-5}$ |
| Pre-Pro | $(1.5 \pm 0.8) \times 10^{-4}$ |
| RNA | $(8.9 \pm 0.6) \times 10^{-4}$ |
| Bunny ($k = 10$) | $(2.6 \pm 0.5) \times 10^{-5}$ |
| Bunny ($k = 50$) | $(2.2 \pm 0.5) \times 10^{-5}$ |
| Bunny ($k = 100$) | $(2.3 \pm 0.5) \times 10^{-5}$ |

| Hyperparameter | Value |
|---|---|
| Layer type | ResNet |
| Residual blocks | 2 |
| Inner depth | 5 |
| Inner width | 512 |
| Activation | ReLU |
| $\beta_R^x$ | 500 |
| $\beta_R^z$ | 0 |
| $\beta_U$ | 10 |
| $\beta_P$ | 10 |
| Latent distribution | uniform |
| Optimizer | Adam |
| Learning rate | $5 \times 10^{-3}$ |
| Scheduler | Exponential w/ $\gamma = 1 - 10^{-5}$ |
| Gradient clipping | 1.0 |
| Weight decay | $3 \times 10^{-5}$ |
| Batch size | 1,024 |
| Step count | 585,600 |
| #Repetitions | 3 |

Table 9: Hyperparameter choices for the rotation experiments. $\beta_U$ and $\beta_P$ are the same for both the sample and latent space.

| Dataset | Number of instances | Noise strength |
|---|---|---|
| Volcano | 827 | 0.008 |
| Earthquake | 6120 | 0.0015 |
| Flood | 4875 | 0.0015 |
| Fire | 12809 | 0.0015 |

Table 10: Dataset overview for the earth data experiments. Each dataset is split into 80% for training, 10% for validation and 10% for testing.

| Hyperparameter | Value |
|---|---|
| Layer type | ResNet |
| residual blocks | 4 |
| Inner depth | 2 |
| Inner width | 256 |
| Activation | sin |
| $\beta_\mathrm{R}^x$ | $10^5$ |
| $\beta_\mathrm{R}^z$ | 0 |
| $\beta_\mathrm{U}$ | $2 \times 10^2$ |
| $\beta_\mathrm{P}$ | 0 |
| Latent distribution | VMF-Mixture ($n_\mathrm{comp} = 5$) |
| Optimizer | Adam |
| Learning rate | $2 \times 10^{-4}$ |
| Scheduler | onecyclelr |
| Gradient clipping | 10.0 |
| Weight decay | $5 \times 10^{-5}$ |
| Batch size | 32 |
| Step count | $\sim 1.2\mathrm{M}$ |
| #Repetitions | 5 |

Table 11: Hyperparameter choices for the earth data experiments. $\beta_\mathrm{U}$ and $\beta_\mathrm{P}$ are the same for both the sample and latent space.

## B.2 Special orthogonal group

To apply manifold free-form flows, we project an output matrix $R \in \mathbb{R}^{3\times3}$ from the encoder/decoder to the subspace of special orthogonal matrices by finding the matrix $Q \in SO(3)$ minimizing the Frobenius norm $\|Q - R\|_F$. This is known as the constrained Procrustes problem and the solution $Q$ can be determined via the SVD $R = U\Sigma V^T$ [Lawrence et al., 2019]:

$$Q = USV^T, \tag{102}$$

where the diagonal entries of $\Sigma$ were sorted from largest to smallest and $S = \mathrm{Diag}(1, \ldots, 1, \det(UV^T))$.

The special orthogonal group data set is synthetically generated. We refer to [De Bortoli et al., 2022] for a description of the data generation process. They use an infinite stream of samples. To emulate this, we generate a data set of $10^7$ samples from their code, of which we reserve 1,000 for validation during training and 5,000 for testing. We vectorize the $3 \times 3$ matrices before passing them into the fully-connected networks. All training details are given in Table 9, one training run takes approximately 7 hours on a NVIDIA A40.

The data set is synthetically generated; of the $N = 100,000$ data points, we use 1% of for validation and hyperparameter selection and 5% for test NLL. Each run uses a different random initialization of papers.

## B.3 Earth data

We follow previous works and use a dataset split of 80% for training, 10% for validation and 10% for testing. For the earth datasets we use a mixture of 5 learnable Von-Mises-Fisher distributions for the target latent distribution. We base our implementation on the `hyperspherical_vae` library [Davidson et al., 2018]. In order to stabilize training we apply a small amount of Gaussian noise to

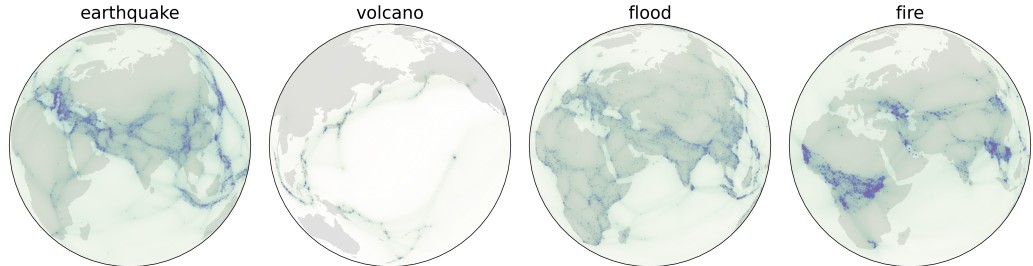

Figure 4: Density estimates of our model on the earth datasets. Blue points show the training dataset, red points the test dataset.

| Dataset | Number of instances | Noise strength |
|---|---|---|
| General | 138208 | 0 |
| Glycine | 13283 | 0 |
| Proline | 7634 | 0 |
| Pre-Proline | 6910 | 0 |
| RNA | 9478 | $1 \times 10^{-2}$ |

Table 12: Details on the torus datasets. Each dataset is randomly split into a train dataset (80%), validation dataset (10%) and test dataset (10%). During training, we add Gaussian noise with mean zero and standard deviation given by 'noise strength' to the data, to counteract overfitting.

every batch (see table 10) and project the resulting data point back onto the sphere. Other training hyperparameters can be found in table 11. Each model trained around 20h on a compute cluster using a single NVIDIA A40.

## B.4 Tori

The torus datasets are randomly split into a train dataset (80%), validation dataset (10%) and test dataset (10%). To counteract overfitting, we augment the RNA dataset with random Gaussian noise. The noise strength and total number of instances is reported in Table 12. We use a uniform latent

| Hyperparameter | Value ($\mathbb{T}^2$) | Value ($\mathbb{T}^7$) |
|---|---|---|
| Layer type | ResNet | ResNet |
| residual blocks | 6 | 2 |
| Inner depth | 3 | 2 |
| Inner width | 256 | 256 |
| Activation | SiLU | SiLU |
| $\beta_{\mathrm{R}}^x$ | 100 | 1000 |
| $\beta_{\mathrm{R}}^z$ | 100 | 100 |
| $\beta_{\mathrm{U}}^x$ | 100 | 100 |
| $\beta_{\mathrm{U}}^z$ | 0 | 1000 |
| $\beta_{\mathrm{P}}$ | 0 | 0 |
| Latent distribution | uniform | uniform |
| Optimizer | Adam | Adam |
| Learning rate | $1 \times 10^{-3}$ | $1 \times 10^{-3}$ |
| Scheduler | oneclyclelr | oneclyclelr |
| Gradient clipping | - | - |
| Weight decay | $1 \times 10^{-3}$ | $1 \times 10^{-3}$ |
| Batch size | 512 | 512 |
| Step count | $\sim 120$k | $\sim 120$k |
| #Repetitions | 5 | |

Table 13: Details on the model architecture, loss weights and optimizer parameters for the torus datasets. We use the same configuration for all protein datasets on $\mathbb{T}^2$.



Figure 5: Log density of M-FFF models in the $(\Phi, \Psi)$-plane of protein backbone dihedral angles (known as Ramachandran plot[Ramachandran et al., 1963]). The learned density matches the true density indicated by the test dataset (*black dots*) very well. Note also that the learned distribution obeys the periodic boundary conditions.

distribution. We train for 120k steps with a batch size of 512 which takes 2.5 to 3 hours on a NVIDIA GeForce RTX 2070 graphics card. Further hyperparameters used in training can be found in Table 13.

### B.5 Hyperbolic space

A straightforward way to define distributions on hyperbolic space (but also other Riemannian manifolds) is, to define a probability density $p_{\text{tangent}}$ in the tangent space at the origin and use the exponential map $\exp_0$ to pushforward this distribution onto the manifold using Eq. (9):

$$\log p_{\text{manifold}}(\exp_0(v)) = \log p_{\text{tangent}}(v) - \log |J_{\exp_0}(v)| - \frac{1}{2} \log \frac{|G_{\text{manifold}}(\exp_0(v))|}{|G_{\text{tangent}}(v)|}, \quad (103)$$

where $G_{\text{manifold}}$ denotes the metric tensor of the embedded manifold and $G_{\text{tangent}}$ the metric tensor of the tangent space. This is also known as a 'wrapped' distribution.

We use the Poincaré ball model, which embeds the n-dimensional hyperbolic space $\mathbb{H}^n$ in the n-dimensional Euclidean space $\mathbb{R}^n$ as defined in Table 2. The exponential map at the origin of this embedding and its Jacobian determinant are simply given by:

$$\exp_0(v) = \tanh(\|v\|) \frac{v}{\|v\|} \qquad \text{and} \qquad |J_{\exp_0}(v)| = \frac{\tanh(\|v\|)}{\|v\| \cosh^2(\|v\|)}. \quad (104)$$

The metric tensor at some point $p \in \mathbb{H}$ is defined by: $G_{\mathbb{H}}^{ij}(p) = \lambda_p^2 \delta_{ij}$ with $\lambda_p = 2/(1 - \|x\|^2)$. The metric tensor of the tangent space is the usual Euclidean metric tensor: $G_{\mathbb{R}}^{ij} = \delta_{ij}$.

With this at hand, we can define latent and toy distributions as wrapped distributions at the origin, as depicted in Fig. 6.

For training, we sample 100k data points from each distribution. Hyperparameters for each model can be found in Table 14. Training takes approximately 2.5, 16, 8 and 16 hours on a NVIDIA A40 graphics card for the one Gaussian, five Gaussians, swish and checkerboard dataset respectively. The resulting model densities are shown in Fig. 6.

### B.6 3D mesh

We base our experiment on the manifold and data provided by [Chen and Lipman, 2024] using 80% for training, 10% for validation and hyperparameter tuning. We report test NLL on the remaining 10% of the data. Each run starts from different parameter initialization. They give the manifold as a triangular mesh, consisting of vertices $v_i \in \mathbb{R}^3, i = 1, \ldots N_v$ and triangular faces $f_j \in \{1, \ldots, N_v\}^3$.

Since the projection to the nearest point on the mesh has zero gradient in parts of $\mathbb{R}^3$, we instead project to the manifold using a separately trained auto encoder with a spherical latent space. This autoencoder consists of an encoder $e : \mathbb{R}^3 \to \mathbb{R}^3$ consisting of five hidden layers with 256 neurons each, SiLU activations and an overall skip connection. The latent codes are computed by projecting the encoder outputs $e(x)$ to a sphere as $z(x) = e(x)/\|e(x)\|$, so that the latent space has the same the topology as the input mesh. Then, a decoder $d(z)$ with the same structure as the encoder is trained to reconstruct the original points by minimizing $\|x - d(e(x)/\|e(x)\|)\|^2$. We train it for $2^{18} = 262,144$ steps, with each batch consisting of all $N_v = 2,502$ vertices and an additional $N = 2,502$ uniformly

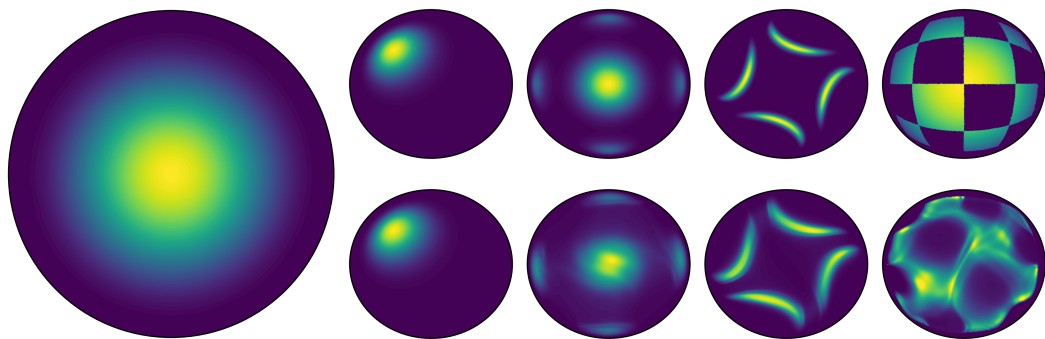

Figure 6: Density estimation on the Poincaré ball model. As latent distribution we use a wrapped normal distribution with standard deviation 0.5 (*Left*). As target distributions (*top row*) we define several toy distributions in the tangent space at the origin and use Eq. (103) to push forward to the manifold. We will reference each distribution from left to right as 'one Gaussian', 'five Gaussians', 'swish' and 'checkerboard'. We train M-FFF on these target distributions using the full expression in Eq. (9) to compute the change in variables and evaluate the densities of the models (*bottom row*). M-FFF are capable to adapt to non-isometrically embedded manifolds. The learned densities on the one Gaussian, five Gaussians and swish dataset closely follow the target densities. On the checkerboard dataset, M-FFF cannot fully reproduce the sharp edges and density of the dataset.

| Hyperparameter | Value (one wrapped) | Value (five gaussians) | Value (swish) | Value (checkerboard) |
|---|---|---|---|---|
| Layer type | ResNet | ResNet | ResNet | ResNet |
| residual blocks | 2 | 6 | 6 | 6 |
| Inner depth | 2 | 3 | 3 | 3 |
| Inner width | 128 | 256 | 256 | 256 |
| Activation | SiLU | SiLU | SiLU | SiLU |
| $\beta_{\mathrm{R}}^{x}$ | 1000 | 1000 | 1000 | 1000 |
| $\beta_{\mathrm{R}}^{z}$ | 100 | 100 | 100 | 100 |
| $\beta_{\mathrm{U}}^{x}$ | 100 | 100 | 100 | 100 |
| $\beta_{\mathrm{U}}^{z}$ | 0 | 0 | 0 | 0 |
| $\beta_{\mathrm{P}}$ | 1 | 1 | 1 | 1 |
| Latent distribution | Wrapped normal | Wrapped normal | Wrapped normal | Wrapped Normal |
| Optimizer | Adam | Adam | Adam | Adam |
| Learning rate | $2 \times 10^{-4}$ | $1 \times 10^{-4}$ | $1 \times 10^{-4}$ | $1 \times 10^{-4}$ |
| Scheduler | Exponential w/ $\gamma = 0.9986$ | onecyclelr | onecyclelr | onecyclelr |
| Gradient clipping | - | - | - | - |
| Weight decay | $1 \times 10^{-3}$ | $1 \times 10^{-3}$ | $1 \times 10^{-3}$ | $1 \times 10^{-3}$ |
| Batch size | 4096 | 4096 | 4096 | 4096 |
| Step count | $\sim 84\mathrm{k}$ | $\sim 485\mathrm{k}$ | $\sim 240\mathrm{k}$ | $\sim 495\mathrm{k}$ |

Table 14: Details on the model architecture, loss weights and optimizer parameters for the Poincaré ball experiments.

random points on the original mesh. We find that for successful training, it is helpful to filter out data with $x_2 > 0.5 + n/10,000$, where $n$ is the step number. This prevents the long bunny ears from collapsing as the are slowly grown, allowing the model to adapt.

We then train M-FFF with the hyperparameters given in table Table 15, using the pretrained autoencoder as our projection to the manifold. Note that we do not train the distribution on the latent sphere of the encoder, but directly on the manifold spanned by it. Training takes approximately 14 hours on a NVIDIA A40.

## B.7 Libraries

We base our code on `PyTorch` [Paszke et al., 2019], PyTorch Lightning [Falcon and The PyTorch Lightning team, 2019], Lightning Trainable [Kühmichel and Draxler, 2023], Numpy [Harris et al., 2020], Matplotlib [Hunter, 2007] for plotting and Pandas [McKinney, 2010, The pandas development team, 2020] for data evaluation. We use the `geomstats` [Miolane et al., 2020, 2023] package for embeddings and projections.

| Hyperparameter | Value |
| --- | --- |
| Layer type | ResNet |
| Residual blocks | 2 |
| Inner depth | 5 |
| Inner width | 512 |
| Activation | ReLU |
| $\beta_{\mathrm{R}}^{x}$ | 1000 |
| $\beta_{\mathrm{R}}^{z}$ | 0 |
| $\beta_{\mathrm{U}}$ | 10 |
| $\beta_{\mathrm{P}}^{x}$ | 100 |
| $\beta_{\mathrm{P}}^{z}$ | 10 |
| Latent distribution | uniform |
| Optimizer | Adam |
| Learning rate | $5 \times 10^{-4}$ |
| Scheduler | Exponential w/ $\gamma = 1 - 0.0039$ |
| Gradient clipping | 1.0 |
| Weight decay | $3 \times 10^{-5}$ |
| Batch size | 1,024 |
| Step count | 469.199 |
| #Repetitions | 3 |

Table 15: Hyperparameter choices for the bunny experiments.

