# OpenReview forum: "Learning Distributions on Manifolds with Free-Form Flows"
_NeurIPS.cc/2024/Conference — NeurIPS 2024 poster_

### Official Review · Reviewer_Z5qx · 2024-06-25

**Soundness:** 2
**Presentation:** 3
**Contribution:** 2
**Rating:** 5
**Confidence:** 4

**Summary:**

In this paper the authors propose to adapt the formalism in Draxtler et al. (2024) and Sorrensen et al. (2024) to Riemannian geometry. The idea they build upon is called Free-form flows and consists of: (i) relaxing the bijectivity requirement of Normalizing Flows by approximating the inverse transformation with a learnable function and (ii) approximating the gradient of the Jacobian determinant with the Hutchinson's trace estimator (and in particular to use the approximate inverse at this point). The contribution consists in generalizing such framework to accommodate for Riemannian manifolds (and specifically to accommodate for the further multiplication by the tangent space orthonormal basis).

**Strengths:**

- the proposed Theorem 3 is elegant and simple, and allows to easily adapt the framework to the manifold under study
- even if not discussed in these terms, the proposed method seem to work particularly well in settings where the underlying density is highly multi modal (e.g. Table 3 and Table 6)

**Weaknesses:**

- even though the authors claim the performance is "mixed" or better "in some cases", the proposed method performs significantly worse than most multi-step approaches across almost all datasets (see Table 4). In almost all other experiments (Table 3, 5, 6) the proposed method performs equal or worse than other multi-step approaches.
- I would then expect the proposed method to highlight a trade-off between performance and computational speed, but no thorough runtime comparison study is performed.

**Questions:**

1. as I understand the method, the main advantage comes from fast inference. Since the performance compared to other methods are worse or equal in most settings (except highly multi modal distributions), I believe it is important to show a runtime comparison with SOTA approaches. In particular, the very recent method in [1] is particularly flexible and much faster than other concurrent work. How does concurrent work compare with the proposed method at fixed performance (e.g. for a given log likelihood value) in terms of runtime (both training and inference)?
2. what are the numerical log likelihood values relative to the plots in Figure 6? how do they compare to other methods?
3. the proposed method relies on two fundamental approximations: estimation of the trace for the Jacobian determinant and approximate inverse of the learnt transformation. In the appendix the authors argue how to reduce the variance by rescaling v and that this helps particularly in low dimensions. I would be interested to see empirically how the variance scales with the dimensions in some experiments
4. given the several losses used in optimization (eq. 22) I was wondering how well is the inverse approximated in practice, which is the second fundamental approximation of the model. Also, the additional two losses kind of suggest that it is hard in practice to learn the approximate inference, which is however fundamental for the method to work. The authors also highlight that the reconstruction error plays a crucial role to learn the correct log likelihood. What are in practice the reconstruction error values (and the other loss terms) in the experiments you have run?

As a general advice, I would suggest the authors to further investigate how and why the proposed approach performs better when the learnt densities are multi modal. I think this is a very promising aspect - and from experience maybe the true limitation of bijective layers - but it is not explored in the paper

[1] Chen et al., Flow Matching on General Geometries, ICLR 2024

[2] Draxler et al., Free-form Flows: Make Any Architecture a Normalizing Flow, AISTATS 2024

**Limitations:**

Limitations are not at all clearly stated as such in the paper. Some limitations are referenced in the checklist but their description is vague and some references are missing (e.g. the checkerboard dataset. Without a comparison is also hard to grasp how much of a limitation this is). I would suggest the authors to include a paragraph concerning limitations, where limitations are mentioned and discussed.

---

> ### Author Rebuttal · Authors · 2024-08-05
>
> We thank the reviewer for the detailed and helpful feedback! Below, we address the points mentioned:
>
> ### No thorough runtime comparison study is performed. [...] How does concurrent work compare [...] at fixed performance [...] in terms of runtime (both training and inference)?
>
> In a nutshell, M-FFF is over an order of magnitude faster at equal inference performance, and computing gradients is cheap even on arbitrary manifolds.
>
> *Inference*
>
> At inference, M-FFF is faster by a factor of typically 100x than competing methods: Wall-clock time is roughly proportional to the number of steps in Tables 3-6, since M-FFF uses approximately the same network sizes as previous methods.
>
> As proposed, we are adding a comparison of M-FFF to Riemannian Flow Matching [1], see Figure 1 in the rebuttal PDF concerning the protein dataset “general”. To speed up R-FM, we reduce the number of steps to sample. We find that **M-FFF is around 50x faster than R-FM at the same NLL.** The single-step performance of R-FM is significantly worse than M-FFF.
>
> In terms of wall-clock time, computing the M-FFF performance in the plot took 30 seconds, the R-FM dot with the same performance took 25 minutes (NVIDIA RTX 2080).
>
>
> *Training*
>
> Overall, computing gradients is cheap for M-FFF, depending on the manifold RFM can be more expensive. In detail: There are two contributions: Neural network evaluations and usage of manifold-related functions (projection, computing tangent spaces, …). In a single training step M-FFF calls the encoder and decoder once each and the projection function twice. R-FM calls its velocity network once, but also needs to integrate the user-specified conditional velocity field, which can be expensive unless closed form expressions are known (e.g. sphere is cheap, the mesh dataset requires solving an ODE at every training step).
>
> In terms of the time to get a model running on a novel manifold, M-FFF simplifies the process as it only requires specifying a projection function. R-FM in addition requires the user to specify a conditional flow monotonously decreasing a premetric, increasing the complexity to set up the model.
>
>
> ### even though the authors claim the performance is "mixed" or better "in some cases", the proposed method performs significantly worse than most multi-step approaches across almost all datasets [...]
>
> M-FFF sets a new SOTA against multi-step approaches in 5 of the 11 experiments presented in Tables 3, 5 and 6 at orders of magnitude faster inference, hence better “in some cases”. With “mixed”, we mean that it outperforms non-SOTA multi-step methods in 3 out of 4 experiments in Table 4.
>
> Among single-step methods, **M-FFF is state-of-the-art in 12 out of 12 experiments**. Notably, it works with little modification across manifolds, where each previous single-step method required special adaptation.
>
>
> ### what are the numerical log likelihood values relative to the plots in Figure 6? how do they compare to other methods?
>
> We will add NLLs to the paper. To the best of our knowledge, there is no established numerical benchmark for learning distributions on hyperbolic space and existing methods use a different embedding. Our experiments confirm that the generalization of our method to non-isometric geometry is correct and yields useful results. We will publish the code to generate our data sets upon acceptance so that future work can use our method as a baseline.
>
>
> ### How [does] the variance scales with the dimensions in some experiments?
>
> From a theoretical point of view, the variance scales just like that of the Riemannian flow matching loss (see below). From an experimental point of view, non-Riemannian free-form flows have been successfully scaled to a 261-dimensional molecular dataset, outperforming previous continuous normalizing flows.
>
> In detail: From appendix D.3 in [1], the variance of the trace estimator $v^T A v$ scales roughly as $2 \\lVert A \\rVert_F^2$. If we apply this to a simple linear model $f(x) = Ax$ and $g(z) = Bz$ we find that summing up the variances of the gradient estimates wrt each element of $A$ leads to a total variance of $2 n \\operatorname{tr}(B^T B)$. A similar calculation for a flow matching loss of the form $\\frac{1}{2} \\lVert Ax - y \\rVert^2$ leads to total variance $n \\lVert x \\rVert^2 \\operatorname{tr}(\\Sigma)$ with $\\Sigma$ the covariance of $p(y|x)$. We can see that both expressions scale explicitly with $n$, with the trace terms also likely scaling with $n$. We thus expect no problems due to variance in high dimensions. We are happy to provide further details on the derivation.
>
> Experimentally, [2] train a Free-Form Flow on a 261-dimensional molecular modeling dataset (QM9). They outperform a previous continuous normalizing flow on this data. Since structurally, M-FFF is very similar to FFF, we expect this behavior to transfer.
>
> - [1] Sorrenson et al., Lifting Architectural Constraints [...], ICLR 2024
> - [2] Draxler et al., Free-form Flows [...] AISTATS 2024
>
> ### What are [...] the reconstruction error values?
>
> They are low, see the table in the rebuttal PDF.
>
> ### Limitations are not at all clearly stated as such in the paper. [...]
>
> We will improve the presentation by adding a dedicated limitations section to the camera ready version. Here are the three main points:
>
> 1. M-FFF approximates the change of variables by replacing the exact inverse with an encoder decoder pair, see l. 190-191, l. 203. This might influence training performance. However, we find that the additional architecture flexibility gained from this approximation lets us outperform concurrent single-step methods with an exact volume change.
> 2. Compared to multi-step methods, M-FFF performs poorly on specific datasets, especially when the density changes sharply or very low density regions are present, see l. 243-245, caption of Figure 6.
> 3. Inaccurate likelihoods when reconstruction error does not drop low enough, see l. 240-242, Appendix B.1

---

> > ### Comment · Reviewer_Z5qx · 2024-08-13
> >
> > I would like to thank the authors for the detailed rebuttal.
> >
> > As reviewers m8ZT and RvPv I was initially concerned about the reconstrcution error and I'm happy to see that in practice it is negligible. I now think about the proposed model as a fast alternative to competing methods and I think it is particularly relevant among single step methods. I still believe, however, that the comparison with multi step methods is not very convincing.
> >
> > I will adjust my score accordingly.

---

### Official Review · Reviewer_nDWB · 2024-07-03

**Soundness:** 4
**Presentation:** 3
**Contribution:** 3
**Rating:** 7
**Confidence:** 5

**Summary:**

This paper introduces M-FFF, a single-step generative model that operates on Riemann manifolds. The authors extend the existing FFF, which functions in Euclidean space, establishing the theoretical foundation for learning distributions on various Riemann manifolds and enabling faster generation than existing multi-step and single-step methods. The effectiveness of the proposed approach is evaluated on several datasets.

**Strengths:**

1. The authors have extended the single-step generator FFF to arbitrary Riemann manifolds, making it possible to generate data in non-Euclidean spaces in a single step with simulator-free which developed in recent flow matching.
2. M-FFF demonstrated superior performance compared to existing flow models in experiments with different datasets. It not only generated high-quality samples but also effectively learned the distribution on each manifold.

**Weaknesses:**

1. Extending existing generative models operating in Euclidean space to Riemannian manifold is a well-trodden path in recent research, so this extension alone lacks novelty. However, the theoretical contributions specific to the extension are solid.
2. The authors evaluated the method's effectiveness only under limited conditions where the dimensionality n is ~10 or less. Given that the theory and method are highly general without specific applications, there should be a discussion on the potential limitations of the algorithm.

**Questions:**

1. Is it realistic to generate data in high-dimensional spaces (e.g., 100 ~ 1000 dimensions), or is it inherently challenging? Understanding the short rebuttal period, I don't expect new experiments, but could you provide insights on the expected impacts of increased dimensionality?
2. Do you plan to release the source code, including reproduction experiments? Considering the scarcity of source code for generative models on Riemann manifolds, releasing the code would significantly contribute to the field.
3. Regarding Specialized architectures in related work, refer to, for example, the paper for quotient manifolds by Yataka et al., Grassmann manifold flows for stable shape generation, NeurIPS2023. While authors of M-FFF claim their theoretical results operate on arbitrary Riemann manifolds, can the same be said for quotient manifolds?
4. M-FFF is described as requiring "requiring only a projection function from an embedding space." However, isn't the calculation of R and Q dependent on the manifold necessary for computing the negative log-likelihood? If so, how are these obtained? At least, it seems there is a discrepancy between the claims in the Introduction and the necessity of Eq. (9) in Section 5 (Toy distributions on hyperbolic space).
5. The authors describe that evaluating the loss in Euclidean space (Eq. (22)) is practically acceptable because "we find that ambient Euclidean distance works well in." However, considering that Riemannian flow matching is rigorously formulated based on the Riemannian metric, this approach seems inconsistent. Theoretically, wouldn’t a formulation using the Riemannian manifold's metric be necessary?
6. Particularly in spaces with negative curvature, such as hyperbolic space, I am concerned about the appropriateness of using Euclidean distance. If some measures are needed, please explain why the formulation in Eq. (22) is appropriate as a general evaluation metric for M-FFF.

**Limitations:**

1. There is no mention of limitations related to dimensionality.

---

> ### Author Rebuttal · Authors · 2024-08-05
>
> We thank the reviewer for the detailed and helpful feedback! Below, we address the points mentioned:
>
> ### Is it realistic to generate data in high-dimensional spaces? [...] There is no mention of limitations related to dimensionality
>
> Yes, it is realistic. From a theoretical point of view, the trace estimator scales similarly to Riemannian flow matching (see below). From an experimental point of view, non-Riemannian free-form flows have been successfully scaled to a 261-dimensional molecular dataset, outperforming previous continuous normalizing flows.
>
> In detail: One metric to assess scaling behavior is to consider the variance of the trace estimator. From appendix D.3 in [1], the variance of the trace estimator $v^T A v$ scales roughly as $2 \\lVert A \\rVert_F^2$. If we apply this to a simple linear model $f(x) = Ax$ and $g(z) = Bz$ we find that summing up the variances of the gradient estimates wrt each element of $A$ leads to a total variance of $2 n \\operatorname{tr}(B^T B)$. A similar calculation for a flow matching loss of the form $\\frac{1}{2} \\lVert Ax - y \\rVert^2$ leads to total variance $n \\lVert x \\rVert^2 \\operatorname{tr}(\\Sigma)$ with $\\Sigma$ the covariance of $p(y|x)$. We can see that both expressions scale explicitly with $n$, with the trace terms also likely scaling with $n$. Since the number of parameters (elements of $A$) scales as $n^2$, the variance per parameter is constant. We thus expect no problems due to variance in high dimensions. We are happy to provide further details on the derivation of these expressions if the reviewer is interested.
>
> Experimentally, [2] train a Free-Form Flow jointly modeling the position and atomic properties (3+6 dimensions) of up to 29 atoms, making this a 261-dimensional dataset (QM9). They outperform a previous continuous normalizing flow on this data. Since structurally, M-FFF is very similar to FFF, we expect this behavior to transfer.
>
> - [1] Sorrenson et al., Lifting Architectural Constraints of Injective Flows, ICLR 2024
> - [2] Draxler et al., Free-form Flows: Make Any Architecture a Normalizing Flow, AISTATS 2024
>
> ### Do you plan to release the source code?
>
> Yes, we will release the source code upon acceptance.
>
> ### While authors of M-FFF claim their theoretical results operate on arbitrary Riemann manifolds, can the same be said for quotient manifolds?
>
> We identify two potential pitfalls when applying M-FFF to quotient manifolds:
>
> 1. Not every quotient manifold is Riemannian. For example the metric must be constant within each equivalence class.
> 2. The topology might change when applying the quotient to an existing manifold, such that potentially a different embedding space and projection needs to be chosen. If the embedding space is the same, the projection must differentiably choose a representative from each equivalence class.
>
> For the Grassman manifold $G(k, D)$, M-FFF can be applied by parameterizing it through idempotent symmetric $k \\times k$ matrices. Projecting to the manifold is then achieved via a singular value decomposition, see e.g. [1].
>
> We think that an extension of M-FFF to arbitrary quotient manifolds is an interesting extension of our work. We thank the reviewer for the interesting pointer to Grassmann Manifold Flows, which we propose to add to our related work.
>
> [1] Implementation of geomstats.geometry.grassmannian.Grassmannian
>
> ### Isn't the calculation of R and Q dependent on the manifold necessary for computing the negative log-likelihood? If so, how are these obtained?
>
> Yes, that is correct. Given a projection $\\pi$, we use that its Jacobian $\\pi’(x) \\in \\mathbb R^m$ spans the tangent space $\\mathcal T_x \\mathcal M$. Then, $Q$ can be obtained by taking any basis of the column space of this Jacobian $\\pi’(x)$. We will make this explicit below Theorem 2.
>
> ### The authors describe that evaluating the loss in Euclidean space (Eq. (22)) is practically acceptable because "we find that ambient Euclidean distance works well in." Theoretically, wouldn’t a formulation using the Riemannian manifold's metric be necessary?
>
> It was not necessary for the cases we considered, but the framework is readily compatible with using geodesic distances. Simply write the reconstruction loss as $\\mathcal L_R = \\mathbb E[d(g_\\phi(f_\\theta(x)), x)^2]$; in the end, both have the same minimum $x =g_\\phi(f_\\theta(x))$. Using Euclidean distance makes computation of the loss cheaper and did not reduce the performance, so we opted for this variant.
>
> ### Particularly in spaces with negative curvature, such as hyperbolic space, I am concerned about the appropriateness of using Euclidean distance
>
> We tried both hyperbolic and the Euclidean distance (in the embedding space). The hyperbolic distance did not give an improvement, so we opted for the Euclidean distance to reduce the computational effort. We find that a low Euclidean reconstruction loss also means a low hyperbolic reconstruction loss (e.g. for the swish experiment, the reconstructions have a Euclidean squared distance of $8 \\times 10^{-6}$ and a squared geodesic distance of $2 \\times 10^{-4}$ from the original samples).
>
> We hope that this addresses the reviewer’s concerns and look forward to the subsequent discussion.

---

> > ### Comment · Reviewer_nDWB · 2024-08-12
> >
> > I appreciate the authors addressing my concerns.
> > - The authors provided the specific limitations on dimensions.
> > - The authors were honest in their responses to my other questions.
> > - I strongly recommend that the authors include the content they provided in the updated version.
> >
> > Overall, thank you for the efforts in addressing my concerns, and I have decided to adjust my recommendation to a score of 7.

---

### Official Review · Reviewer_RvPv · 2024-07-16

**Soundness:** 3
**Presentation:** 3
**Contribution:** 3
**Rating:** 7
**Confidence:** 5

**Summary:**

The extends a recent class of generative models, called free-form flows, to Riemannian manifolds.  These models are trained like normalizing flows, but relax the bijective constraint and instead employ separate encoder and decoder networks that are trained to be inverses of each other.  The proposed method differs from existing flows for manifolds in that it constructs a map from the manifold M, back to M, using a full rank transformation plus a projection.  This differs from existing manifold flows which typically use injective transformations.  The authors derive a change of variables on manifolds for this setting and derive an unbiased estimator of its gradient and propose a loss function that affords stable training of these models.  The method consistently outperforms existing generative models on manifolds at learning the various target distributions from samples at a significantly lower expense than previous methods.

**Strengths:**

- The paper is well written
- The idea is novel and extends an interesting line of work around a new class of generative models
- The method seems to perform well in practice, although I have some concerns
- Section 3 includes a loss function that mitigates potential failure cases like the networks failing in low data regions, or the ambient space encoder mapping to points away from the manifold.
- The method is evaluated on the typical datasets seen in flows for manifold papers.

**Weaknesses:**

- I'm not fully convinced by the likelihood based evaluation.  The invertibility of normalizing flows ensures that the flow can produce samples anywhere on $\mathcal{M}$ and similarly, can assign a likelihood to any point on $\mathcal{M}$.  Since free form flows are not invertible, it is possible that the flow does not map to all of $\mathcal{M}$.  This would make the comparison between the likelihood values of normalizing flows and free form flows invalid because the likelihoods are valid on different domains, which could potentially invalidate the reported results.  Instead, the experiments could report a sample based metric to measure the method's performance.
- There is no error bound on the likelihood gradient like there is in Draxler et al. 2024.  Although it seems like the method performs well in practice, I would like to know exactly how much is lost by using a decoder network instead of a true inverse.  In injective flows, I don't think that the free form flow gradient works because $\nabla \log |f'(x)| = \text{Tr}{(\nabla f'(x){f^+}'(z))}$ where ${f^+}'$ is the pseudo inverse of $f'$, so using any other possible inverse would give a wrong gradient.  I understand that the proposed method is not an injective flow and might not suffer from this, but I would like to see a gradient error bound to know for sure.
- The $m$ and $n$ dimensions in the proof of theorem 2 seem to be flipped in the appendix.

**Questions:**

- It would be nice to clarify in the text that all of the quantities considered are written in terms of the Euclidean ambient space coordinates.

**Limitations:**

The authors adequately addressed the limitations.

---

> ### Author Rebuttal · Authors · 2024-08-05
>
> We thank the reviewer for the detailed and helpful feedback! Below, we address the points mentioned:
>
> ### I'm not fully convinced by the likelihood based evaluation. [...] Since free form flows are not invertible, it is possible that the flow does not map to all of $M$
>
> Yes, this could happen in principle if the reconstruction loss is not fulfilled. To prevent this pathology, we monitor the reconstruction loss of validation data during training. In the rebuttal PDF, we report the reconstruction loss after training for all datasets, which are consistently low.
>
> We will add these numbers and the above explanation to the main paper.
>
>
> ### There is no error bound on the likelihood gradient
>
> Good point! We find that the same derivation applies to M-FFF, only that the Jacobians are projected to the tangent space. Here’s the bound we find:
>
> $ \\left| \\text{tr}(R^T (\\nabla_{\\theta} J_{f_\\theta}) J_{g_\\phi} R) - \\nabla_{\\theta} \\log |R^T J_{f_\\theta} Q| \\right| \\leq \\lVert R^T (\\nabla_{\\theta} J_{f_\\theta}) J_{f_\\theta^{-1}} R \\rVert_F^2 \\lVert R^T J_{f_\\theta} J_{g_\\phi} R - I_n \\rVert_F^2 $
>
> Notation: $J_{f_\\theta} = f_\\theta'(x)$, $J_{g_\\phi} = g_\\phi'(z)$ and $J_{f_\\theta^{-1}} = f_\\theta^{-1\\prime}(z)$. We provide a formal statement in the rebuttal PDF, and will add a proof to the camera ready version (if the reviewer is interested, we can also post it here).
>
> ### The $m$ and $n$ dimensions in the proof of theorem 2 seem to be flipped in the appendix. [...] It would be nice to clarify in the text that all of the quantities considered are written in terms of the Euclidean ambient space coordinates.
>
> The dimensions are not flipped, but we did not make the use of internal/ambient coordinates explicit enough. We will clarify this as needed for the camera ready version.
>
> We hope that this addresses the reviewer’s concerns and look forward to the subsequent discussion.

---

> > ### Comment · Reviewer_RvPv · 2024-08-08
> >
> > Thank you for the response and for clearing up some of my concerns.  I appreciate adding reconstruction losses to the text however I am still concerned about using comparing likelihoods from the proposed model to those produced by exact flows.  My concern stems from the possibility that if the model does not map to the entire data space, then its values for probability density can be inflated compared to a model that maps to the entire data space.  For example, a uniform distribution over a space with volume $V$ has a density of $\frac{1}{V}$, but a uniform distribution over a subset of the region, with volume $V - T$ where $T>0$, has a larger density of $\frac{1}{V - T}$.  It is possible that something similar is happening in your experiments which is why I would like to see sample based evaluation.  Please let me know if you do not agree with this reasoning.

---

> > > ### Author Response · Authors · 2024-08-12
> > >
> > > Thank you for the clarification and the continued interest! We do think that the reconstruction losses validate reporting likelihoods. Below, we provide more details why reconstruction errors are useful to detect the described behavior, as well as complimentary experimental evidence that no important regions are left out by M-FFF.
> > >
> > > Let us consider the reviewer’s example of a uniform distribution, as this is an instructive case. Suppose that the model learns a uniform distribution, but leaves out a region $T$. Then we agree, it would result in a spurious likelihood. However, the reconstruction loss notices: A point in $T$ is moved by the encoder-decoder pair into $V \\setminus T$ (since the generator only produces points on $V \\setminus T$). Thus, all points in $T$ are moved into $V \\setminus T$, increasing reconstruction loss. The reconstruction losses consistently lower than 1 by three or more orders of magnitude indicate that no such left-out region exists or it is negligibly small.
> > >
> > > Experimentally, we additionally verify that **every test data point has a generated point close by** on all $n=2$ manifolds (sphere, bunny, torus). To test this, we divide the manifold into bins and confirm that in each bin with a test data point, M-FFF also generates samples. We test this by sampling often enough from the trained model, until there is a sample in each bin. We choose natural bins for each manifold (icosphere, subdivision of original trimesh, rectangular grid of polar representation) and make the bins small (bin area $1/100,000$ compared to total manifold area). We confirm for every run for every $n=2$ experiment that in every bin populated by test data M-FFF also generates samples.
> > >
> > > The problem with a sample-based metric is that competitors have not made their pretrained models available, so a thorough comparison would not be possible. We propose adding sample-based evaluations to the camera ready revision for future work to compare against. As a preliminary metric, we report the Wasserstein-2 distance for the $n=2$ datasets above:
> > > | Dataset | Wasserstein-2 distance |
> > > |---------------|---------------|
> > > | Bunny (K=10)  | 0.09 ± 0.02   |
> > > | Bunny (K=50)  | 0.046 ± 0.006 |
> > > | Bunny (K=100) | 0.026 ± 0.007 |
> > > | General       | 0.21 ± 0.04   |
> > > | Glycine       | 0.32 ± 0.05   |
> > > | Proline       | 0.51 ± 0.05   |
> > > | PrePro        | 0.47 ± 0.04   |
> > > | Flood         | 0.047 ± 0.010 |
> > > | Fire          | 0.072 ± 0.027 |
> > > | Volcano       | 0.249 ± 0.035 |
> > > | Earthquake    | 0.068 ± 0.022 |

---

> > > > ### Comment · Reviewer_RvPv · 2024-08-12
> > > >
> > > > Thank you for the response, I agree that small reconstruction errors might mitigate the likelihood issue and think that identifying how they are related is out of the scope of this paper.  Also thank you for reporting the W2 distance of your method on the various datasets, this is a great item for future works to compare against.  Since you have addressed all of my concerns, I'll update my score.

---

### Official Review · Reviewer_m8ZT · 2024-07-16

**Soundness:** 4
**Presentation:** 4
**Contribution:** 3
**Rating:** 6
**Confidence:** 4

**Summary:**

This paper introduces Manifold Free-Form Flows (M-FFF), an extension of free-form flows to Riemannian manifolds. M-FFF is easy to adapt to an arbitrary Riemannian manifold, since the method requires only a projection function from an embedding space. Moreover, the method itself relies only on a single function evaluation during training and sampling, which speeds up inference over multi-step methods by several orders of magnitude. As demonstrated by the experimental section, M-FFF consistently matches or outperforms single-step methods on several benchmarks on spheres, tori, rotation matrices, and hyperbolic space, while being reasonably competitive with multi-step methods.

**Strengths:**

1. This adaptation of free-form flows to manifolds is novel and allows for unconstrained flows over manifolds that are easier to learn (due to fewer geometric restrictions) and much faster as a consequence of the single-step nature of the method.

2. The results given are quite reasonable and show that M-FFF is an excellent single step method and competitive with multi-step methods, while being much faster.

**Weaknesses:**

1. The method, in essence, trades the hard manifold constraints and theoretical cleanliness of previous Riemannian Normalizing Flows (Mathieu et al., 2020) for a projection-based approximation. Specifically, the benefit of a loose and adaptable "encoder-decoder" framework comes at the cost of having to optimize three additional losses: the reconstruction loss $\mathcal{L}_R$, the uniformity loss $\mathcal{L}_U$, and the projection loss $\mathcal{L}_P$. That is to say, unless these losses are optimized well, the geometry may be respected very approximately, which is a drawback relative to existing multi-step methods. The authors themselves admit that they have had issues with the reconstruction loss being too high (see Section 5). Additionally, such a method may be rather suboptimal for applications where there are hard physical constraints, e.g. see learning symmetric densities over $SU(n)$ [a] in the context of lattice QFT for downstream use in gauge integrals. M-FFF is nonetheless an interesting method, but these limitations should be well-noted.

2. There are clearly contexts in which this method is suboptimal relative to existing multi-step methods, e.g. see Figure 6. Due to the many approximations made in the method, it can be hard to match some of the more rigid density geometry exhibited by the checkerboard pattern, that strict on-manifold multi-step methods have no trouble learning (see Lou et al. (2020)). This should also be well noted; the speed flexibility of M-FFF come at a cost.

### References

[a] Equivariant manifold flows. https://arxiv.org/abs/2107.08596

**Questions:**

My questions to the authors are listed below:

1. In Section 5, the authors mention that the "reconstruction error sometimes does not drop to a satisfactory level." My question is, when does this usually happen, and can we more precisely characterize the situations in which one of the non-NLL losses fails to give a satisfactory result? I believe this question is key to better determining when precisely M-FFF is a reasonable and/or good alternative to strict multi-step methods such as Lou et al. (2020) or Mathieu et al. (2020).

### Additional Comments and Minor Corrections

The writing in this paper is for the most part quite clear and the presentation of material is excellent, with several expository figures given to elucidate important parts of the method. Nonetheless, I would like to give a non-exhaustive list of minor corrections below:

L62: "for building normalizing flows such as SO(3)" -> "for building normalizing flows for manifolds such as SO(3)"

L203: I believe the authors meant to say the substitution happens for equation (13), and the results is shown in equation (19). That is, "This allows us... in Eq. (19)" -> "This allows us.... in Eq. (13)"

L228: "by rotations matrices with positive determinant" -> "by rotation matrices with positive determinant"

**Limitations:**

Although some limitations of the method are mentioned throughout the paper, this work would benefit from an explicit section made to address limitations of the method (in particular incorporating some of what I have mentioned in the "Weaknesses" section above).

---

> ### Author Rebuttal · Authors · 2024-08-05
>
> We thank the reviewer for the detailed and helpful feedback! Below, we address the points mentioned:
>
> ### The geometry may be respected very approximately
>
> No, this is a misunderstanding. M-FFFs always respect the geometry: In Eq. (17), we exactly project the output of an unconstrained neural network to the manifold. Thus, encoder and decoder are guaranteed to be manifold-to-manifold functions via the projection $\\pi : \\mathbb R^m \\to \\mathcal M$ (see Fig 1, right).
>
> The projection loss encourages the unconstrained neural network to produce points close to the manifold before they are projected to the manifold. Therefore, the projection only leads to small (and thus numerically stable) corrections, and topologically problematic mappings (e.g. to the center of a Euclidean embedded sphere) are avoided. The uniformity loss encourages encoder and decoder to become inverses of each other even at locations where only few training points are available, i.e. where the NLL training signal is weak.
>
> Thus, projection and uniformity loss mainly complement the NLL loss, and we generally find that M-FFFs are not very sensitive to the relative strength of these losses, see the ablation in Figure 1b provided in the rebuttal pdf.
>
>
> ### learning symmetric densities over $SU(n)$
>
> We need to disambiguate two requirements:
>
> 1. Distributions that are *invariant* under the action of  $SU(n)$.
> 2. Distributions that assign a probability to the *elements* of $SU(n)$.
>
> The reviewer’s remark refers to requirement 1, whereas our paper addresses requirement 2: M-FFFs can exactly learn distributions over the elements of $SU(n)$ by embedding $U \\in SU(n)$ into $\\mathbb C^{n \\times n}$, and associating this space with $\\mathbb R^{2 \\times n \\times n}$. We can project from $\\mathbb C^{n \\times n}$ to the $SU(n)$ manifold by various methods, e.g., via polar decomposition or via the constrained Procrustes algorithm.
>
> We agree that combining density estimation on manifolds with group invariance (requirement 1) is an interesting future direction.
>
>
> ### There are clearly contexts in which this method is suboptimal relative to existing multi-step methods [...] This should also be well noted
>
> Multi-step and single-step methods occupy different spots in the trade-off between competing modeling goals: Multi-step methods have slightly higher generative quality, especially in the presence of sharp boundaries in the density. Single-step methods are $\approx 100\times$ faster and allow for much more efficient explicit calculation of the probability density (expensive integration of the instantaneous change-of-variables is avoided). Note also that our method outperforms previous single-step approaches, even when they are specialized to a specific type of manifold. See our response to all reviewers above for more details.
>
> As an example, Figure 1 in the rebuttal PDF compares the performance of M-FFF to Riemannian Flow Matching: R-FM requires 50x the compute to achieve the quality of M-FFF on the protein dataset “General”.
>
> ### In Section 5, the authors mention that the "reconstruction error sometimes does not drop to a satisfactory level." My question is, when does this usually happen, and can we more precisely characterize the situations in which one of the non-NLL losses fails to give a satisfactory result?
>
> Yes, we are happy to elaborate. In general the likelihoods we measure with M-FFF are trustworthy as long as the validation reconstruction errors are low. This is the case for all datasets as shown in our rebuttal pdf. However, if there are sharp edges present in the probability density, a small error in reconstruction near the edge can have a large effect on the measured likelihood. We identify this problem in the earth datasets (e.g. many floods at coastlines, but none at sea) and present a diagnostic to verify the measured likelihoods in appendix B.1.
>
> We will add this explanation to Section 5 and will add it to an explicit limitation section in the camera ready version.
>
> We hope that this addresses the reviewer’s concerns and look forward to the subsequent discussion. We thank the reviewer for the list of corrections which we will incorporate into the camera ready version.

---

> > ### Comment · Reviewer_m8ZT · 2024-08-11
> > **Thank you for the rebuttal**
> >
> > Thank you for your rebuttal comments; in particular, the comments regarding when reconstruction error does not drop to a satisfactory level. I maintain my original score and believe this paper should be accepted.

---

### Author Rebuttal · Authors · 2024-08-05

We thank all reviewers for their helpful and constructive feedback. Below, we collect and address the main concerns by the reviewers.



### Performance comparison (reviewers m8ZT, Z5qx)

Among single-step methods, **M-FFF is state-of-the-art in 12 out of 12 experiments**. Notably, it works with little modification across manifolds, where each previous single-step method required special adaptation.

Multi-step methods occupy a different spot in the trade-off between competing modeling goals: They have slightly higher generative quality, especially in the presence of sharp boundaries in the density, which comes at the cost of $\approx 100\times$ slower sampling. Despite this, **M-FFF outperforms SOTA multi-step methods in 5 of the 15 experiments** we perform.

To evaluate the performance of multi-step methods with varying compute, we reduce the sampling steps of a Riemannian Flow Matching model trained on protein backbone angle data “general”. We find **that M-FFF is 50x faster than Riemannian Flow Matching [1] at the same quality**.

### Scaling with dimension (reviewers Z5qx, nDWB)

We do not foresee a problem with scaling M-FFF to manifolds with higher dimension given the theoretical scaling of the trace estimator with dimension and the structural similarity to free-form flows [2].

### Reconstruction losses (reviewers Z5qx, m8ZT, RvPv)

We confirm that validation reconstruction losses are low in the rebuttal pdf. We will add these numbers to the camera ready revision.

### FFF error bound transfers to M-FFF (reviewer RvPv)

We confirm that the error bound on the gradient of the likelihood as derived in [2] transfers to the surrogate as proposed for manifold free-form flows (see Theorem 1 in the rebuttal pdf). We propose to add this theorem and a proof in the appendix of the paper.

[1] Chen et al., Flow Matching on General Geometries, ICLR 2024
[2] Draxler et al., Free-form Flows: Make Any Architecture a Normalizing Flow, AISTATS 2024

We give more details on all points in the respective answer to the reviewers and look forward to the subsequent discussion.

---

### Decision · Program_Chairs · 2024-09-25

**Decision:**

Accept (poster)

**Comment:**

The provides a simple extension of free-form flows to manifolds. While the work is incremental (the driving theory was available in the recent papers by Draxler & Sorrenson) it nonetheless makes the ideas explicit and the empirical demonstrations are valuable. The authors are strongly encouraged to update the paper according to the key discussions, in particular, the low reconstruction loss was generally deemed an important concern. The main empirical observation is the improvement of single-step methods; it would perhaps be reasonable to tone down the claims regarding the comparison to multi-step methods as the submitted text could be misunderstood to imply that the proposed method is competitive with such.